# TOAM-YOLO: A Tiny Object-Aware Multi-Expert YOLO Framework for Diverse Domains

**Vaibhav Sharma** *  
*SBILab, IIIT-Delhi, India*  
*Department of Onco-Hematopathology, AIIMS-New Delhi, India*

*vaibhavs@iiitd.ac.in*

**Arnesh Batra** *  
*SBILab, IIIT-Delhi, India*

*arnesh23129@iiitd.ac.in*

**Arush Gumber** *  
*SBILab, IIIT-Delhi, India*

*arush23136@iiitd.ac.in*

**Ritu Gupta**  
*Department of Onco-Hematopathology, AIIMS-New Delhi, India*

*drritugupta@gmail.com*

**Anubha Gupta**  
*SBILab, IIIT-Delhi, India*

*anubha@iiitd.ac.in*

**Reviewed on OpenReview:** *https://openreview.net/forum?id=2lIE1tgmRN*

## Abstract

YOLO-based object detection models have advanced significantly over the years through continuous architectural refinements and subsequent performance improvements. Tiny object detection remains a challenging task due to several constraints posed by progressive downsampling in model architectures and the smaller footprint of tiny objects in high-resolution images. This challenge is faced in diverse applications such as maritime surveillance, aerial surveillance, and in medical applications such as microscopic blood cell analysis. In this study, we introduce and demonstrate that a novel multi-domain expert, which we refer to as TOA-MoE (Tiny object aware mixture of experts), consisting of a Hessian-based curvature expert and a Fourier-based frequency expert, along with a 3-level attention mechanism, substantially improves the detection performance of YOLO models while only increasing the learnable parameters by a small fraction. Additionally, we add a high-resolution P2-level detection pathway integrated with a feature fusion network that incorporates a BiFPN-style structure and integrates deformable convolutional layer modules in the architecture, and we replace the standard up-sampling layers with a Content-Aware Reassembly of Features (CARAFE) module to preserve fine-grained feature details during feature map expansion. We systematically demonstrate the plug-and-play capability of these changes on YOLOv11 and YOLOv12 models. Tiny object aware Mixture of experts-based YOLO (TOAM-YOLO) achieves consistent improvements across five datasets: three tiny object benchmarks (SeaPerson, TinyPerson, VisDrone) with mAP@0.5 improvements of 11.6%, 4.64%, and 11% respectively, and two blood cell datasets (BCCD, CBC) with mAP@0.5:0.95 improvements of 3.8% and 1.7% for platelet detection, all over YOLOv12n, while adding only 0.75M parameters. Ablation studies show that the high-resolution P2-level detection pathway provides the largest performance gain, with TOA-MoE providing an additional improvement on top.

---

*Equal contribution

# 1 Introduction

The challenge to precisely represent and localize objects with a small spatial footprint within high-resolution images is the fundamental reason why tiny object detection is still a persistent challenge in both the natural and biomedical domains Yao et al. (2025); Yang et al. (2025). In general-purpose settings such as surveillance, aerial imaging, and autonomous navigation, objects like pedestrians or vehicles at a distance appear with low signal-to-noise ratios in images. They are often indistinguishable from background clutter after multiple stages of convolutional downsampling.

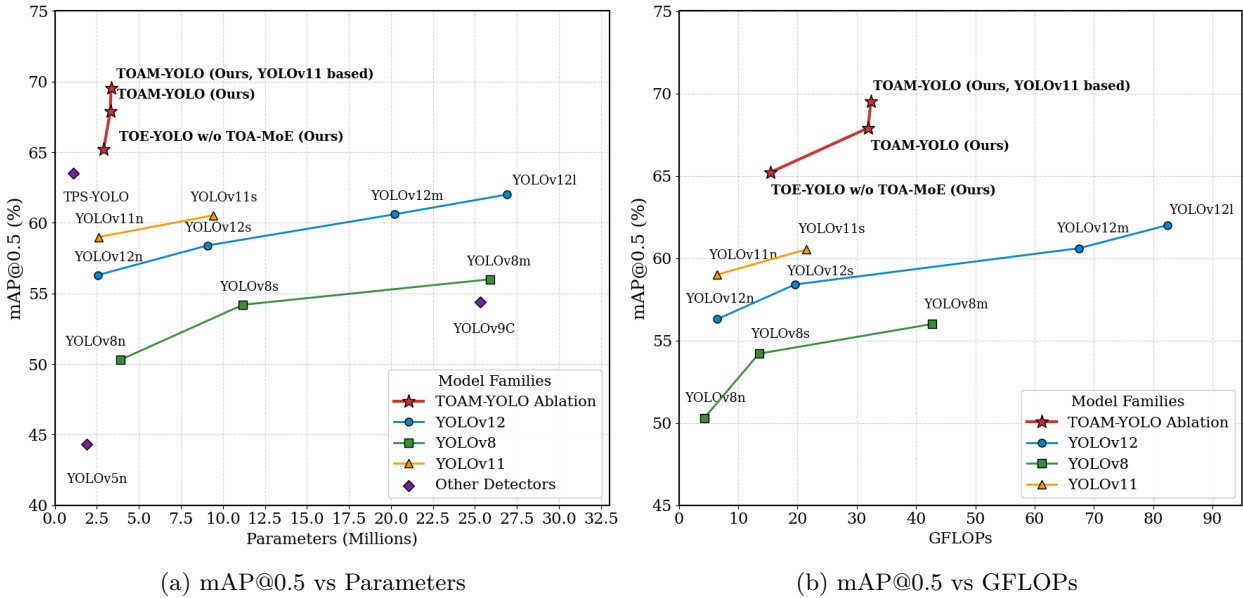

(a) mAP@0.5 vs Parameters       (b) mAP@0.5 vs GFLOPs

Figure 1: Trade-off analysis of mAP@0.5 vs. Parameters (left) and mAP@0.5 vs. GFLOPs (right) on the SeaPerson dataset, comparing TOAM-YOLO, TOE-YOLO and TOAM-YOLO (YOLOv11 based) with YOLOv8, YOLOv11 and YOLOv12 model families, showing that TOAM-YOLO outperforms larger and more complex YOLO models while having only 3.3M Parameters.

Similarly, in medical imaging applications, such as automated blood cell detection, red blood cells (RBCs), white blood cells (WBCs), and platelets frequently manifest as low-resolution instances with minimal pixel coverage, leading to significant feature degradation and poor discriminability. Imaging artifacts including blur, occlusion, deformation, and viewpoint distortions compound the difficulty of extracting stable representations Yang et al. (2025).

This motivates approaches that combine local feature extraction with global attention mechanisms, such as transformers, to capture both fine-grained details and long-range dependencies Chen & Lu (2025); Muzammul & Li (2025). In digital pathology, quantitative or qualitative alterations in blood cells such as changes in count, size, shape, or color serve as biomarkers for a vast spectrum of medical conditions, including anemia, leukemia, and various infections Putzu et al. (2014); National Academies of Sciences, Engineering et al. (2023).

In medical, satellite and drone images, informative elements often manifest as classic tiny objects spanning less than 20×20 pixels Tyagi et al. (2023). You Only Look Once (YOLO) models have revolutionized object detection over the past decade, yet their application to tiny objects across various domains including medical imaging, still presents significant hurdles. The hierarchical architecture of CNNs is meant to progressively extract abstract, high-level semantic features, but consequently suffer from a progressive diminishment of spatial resolution through successive down-sampling layers. This leads to a loss of fine-grained information, indispensable for an accurate localization and classification of tiny objects, especially when they appear in dense, overlapping clusters, as often seen with blood cells in medical images or small components in industrial inspection, or distant vehicles in surveillance Sapkota et al. (2025).

YOLOv12 Tian et al. (2025) replaces the CNN stem of YOLO model architecture with an attention-centric design using area attention and Residual Efficient Layer Aggregation Networks (R-ELAN), along with flash attention Dao et al. (2022) for improved performance and efficiency. Building on this, we propose TOAM-YOLO, an improved YOLOv12n variant with only 3.3M parameters that surpasses larger YOLO models and SOTA methods across multiple domains. It achieves top performance on tiny object datasets like SeaPerson Yu et al. (2022), TinyPerson Yu et al. (2020), VisDrone Zhu et al. (2021), and extends well to medical datasets such as BCCD Shenggan (2019) and CBC Alam & Islam (2019). Figure 1 presents a comparison of the mAP@0.5 achieved by TOAM-YOLO against baseline YOLO model families on SeaPerson Dataset with respect to the number of parameters (left) and GFLOPs (right). Figure 1b illustrates the trade-off between computational complexity (GFLOPs) and detection accuracy relative to the YOLOv12n baseline. While the proposed approach incurs a higher computational cost, this increase is compensated by a marked improvement in detection accuracy, especially for tiny objects that are often inadequately captured by lightweight architectures. Key contributions of this study are as follows:

- We introduce a novel 2-stage module, namely 'Tiny Object Attention with Mixture-of-Experts' (TOA-MoE). It uses multiple experts specializing in spatial, frequency, and curvature (Hessian-based) domains. Subsequently, it refines these features with a multi-granularity attention hierarchy operating at the pixel, patch, and region levels.

- We add a high-resolution detection head specifically for tiny objects, integrating our TOA-MoE module for improving small-scale target detection and resolvability.

- We have designed a more powerful feature fusion network. This includes a structure inspired by the Bi-directional Feature Pyramid Network (BiFPN) Chen et al. (2021) for multiscale information blending from different layers. It also incorporates Deformable Convolutions (DCNv3)Dai et al. (2017) to allow better adaptability to irregular object shapes.

- Instead of using simple traditional upscaling methods, we use Content-Aware Reassembly of Features (CARAFE)Wang et al. (2019). This smarter method rebuilds fine-grained details when increasing the resolution of feature maps, preventing important information about tiny objects from being lost.

## 2 Related Work

### 2.1 Evolution of Real-Time Object Detectors

Real-time, single-stage object detectors within the YOLO family have long dominated applications requiring both speed and accuracy. From the original YOLO Redmon et al. (2016) to the current YOLOv12 Tian et al. (2025), successive iterations have introduced anchor-based regression, CSP-style backbones, multi-scale detection heads, and attention-driven enhancements, enabling SOTA object detection under real-time constraints. YOLOv12-N achieves 40.6% mAP on COCO with fewer FLOPs and parameters.

### 2.2 Transformer-based frameworks

Beyond YOLO and hybrid CNN-transformers, purely transformer-based frameworks, such as the DETR (Detection Transformer) Carion et al. (2020) family are also popular. The original DETR introduced an end-to-end prediction paradigm, eliminating the need for anchor design and non-maximum suppression by treating object detection as a set prediction problem. While powerful, DETR suffered from slow convergence and limited performance on small or tiny objects due to the low spatial resolution of query-feature representations. Variants of DETR such as Deformable DETR Zhu et al. (2020) and RT-DETR Zhao et al. (2024) improve convergence and efficiency, even surpassing YOLO models in speed and accuracy, though they remain weak at tiny object detection as indicated by our results in Table 5.

### 2.3 Advances in Tiny Object Detection

Tiny object detection remains a core challenge, especially in medical imaging, as CNNs downsample feature maps, causing loss of critical spatial detail. Transformers like DETR also inherit this limitation through

CNN backbones that reduce feature resolution early. To address this, TPS-YOLO Yao et al. (2025), built on YOLOv8, improves detection in occluded, long-range scenes by preserving shallow P2 features, using their SPDCA modules, and applying lightweight depth-wise convolutions. It maintains high accuracy on SeaPerson Yu et al. (2022) and VisDrone-person Zhu et al. (2021) datasets, while remaining efficient for edge deployment. DDH-YOLO Liu & Yang (2025) improves small object detection in UAV aerial imagery using a dual-detection-head and, C2f-DD and PFPN modules for enhanced feature extraction achieving a mAP@0.5 of 40.4% on the VisDrone dataset. CST-YOLO Kang et al. (2024) combines CNNs with Swin Transformers to enhance small object detection, achieving 92.6% mAP@0.5 on BCCD Shenggan (2019) and CBC Alam & Islam (2019) datasets outperforming YOLOv7, but at a very high computational cost (47.5M parameters, 235 GFLOPs).

## 3 Methodology

We present our proposed object detector, namely, TOAM-YOLO (Tiny Object Attention with Mixture-of-Experts YOLO). We consider the robust YOLOv12 Tian et al. (2025) framework and introduce modifications to the feature fusion neck, culminating in a novel attention module specifically designed to enhance the detection of tiny objects.

### 3.1 Architectural Overview

Our architecture (Figure 2) builds upon YOLOv12 by adding a high-resolution **P2-level detection head**, branching early from the backbone. While standard YOLOv12 detects at P3–P5 levels, our P2 head retains fine-grained spatial detail and, as shown in the progressive ablations (Table 4), contributes the most to tiny-object sensitivity. This path is refined using our **Tiny Object Attention with Mixture-of-Experts (TOA-MoE)** module and Area attention blocks (A2C2f) before prediction. TOA-MoE is strategically placed in the P2 feature path to further refine features for the smallest targets Hu et al. (2018); Woo et al. (2018), while CARAFE Wang et al. (2019) supports detail-preserving upsampling along this pathway. To improve detection across all scales, we introduce a BiFPN-style Tan et al. (2020) cross-scale connection that enriches the P3 feature map by fusing outputs from both the backbone and the top-down path.

Furthermore, Deformable Convolutions v3 (DCNv3) Dai et al. (2017); Wang et al. (2023) are employed in the bottom-up path to provide flexibility to the model in adapting to the morphological variance of objects like blood cells. We refer to the YOLOv12n model integrated with only the proposed architectural enhancements, excluding the TOA-MoE module as **Tiny Object Enhanced YOLO (TOE-YOLO)** for ablation studies. In essence, TOE-YOLO represents the TOAM-YOLO framework without the TOA-MoE module.

### 3.2 The Proposed TOA-MoE Module

The central innovation of our work is the TOA-MoE module (Figure 3), a two-stage attention mechanism designed to overcome the limitations of standard attention in the context of tiny object detection. We posit that accurate identification of tiny objects requires a comprehensive understanding derived from multiple feature domains and granularities. Figure 4 further illustrates that the two experts exhibit complementary specialization across both datasets, validating our core design motivation. It consists of the following two stages that collectively address the multi-faceted challenge of tiny object detection:

1. **Multi-Domain Feature Enrichment:** A Mixture-of-Experts (MoE) front-end dynamically enriches the input features by routing them through three parallel expert streams: a standard spatial stream, a frequency-domain stream, and a curvature-based stream.

2. **Multi-Granularity Hierarchical Attention:** The enriched features are then processed by a 2nd MoE-gated, hierarchical attention that simultaneously analyzes information at the pixel, patch, and region levels.

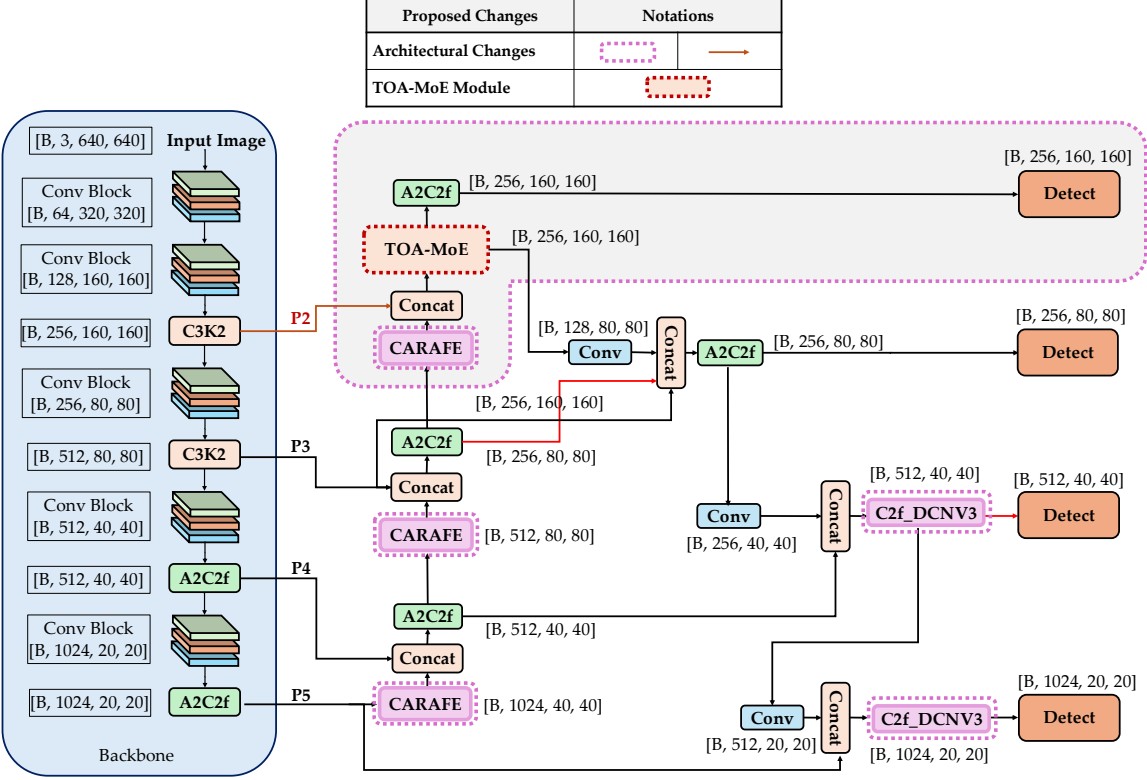

Figure 2: Architecture diagram of TOAM-YOLO with architectural modifications as described in the legend. Our design extends the YOLOv12 backbone with a feature fusion neck tailored for tiny objects. We add a high-resolution **P2 detection head** for improved sensitivity to small targets, powered by the novel **TOA-MoE module** (see Figure 3). In the top-down path, standard upsampling is replaced by three **CARAFE** operators. The bottom-up path uses a **BiFPN-style connection** for better fusion and includes two **C2f-DCNV3** blocks to handle high morphological variance.

### 3.2.1 Stage 1: MoE for Multi-Domain Feature Enrichment.

Recognizing that tiny objects can be characterized by more than just their local spatial patterns, we first aim to create a maximally informative feature space.

Our module processes the input tensor $X \in \mathbb{R}^{B \times C \times H \times W}$ through an MoE front-end, where each expert specializes in extracting a complementary type of information.

- **Spatial Expert ($E_{id}$):** This pathway employs an identity mapping, $E_{id}(X) = X$, which preserves the original high-fidelity spatial features.

- **Frequency Expert ($E_{freq}$):** To understand the global distribution and periodicity of tiny objects Wang et al. (2025b); Sun et al. (2025), this expert leverages the 2D FFT ($\mathcal{F}$), inspired by Fourier-based attention networks Chaudhuri et al. (2024). Frequency domain processing allows for an efficient, full-image receptive field to capture global contextual patterns, such as the texture of a blood smear, that are computationally prohibitive for standard convolutions and help disambiguate individual objects from background patterns. It captures full-scale global information by utilizing FFT.

- **Curvature Expert ($E_{hess}$):** Tiny objects often manifest visually as intensity "blobs." Motivated by Hessian-based blob detection algorithms Marsh et al. (2018); Liu et al. (2010); Xu (2014), this expert is designed to identify such structures by analyzing local surface curvature. It fuses the multi-scale 2nd-order partial derivatives ($f_{xx}, f_{yy}, f_{xy}$) with the Determinant of Hessian ($DoH = f_{xx}f_{yy} - f_{xy}^2$),

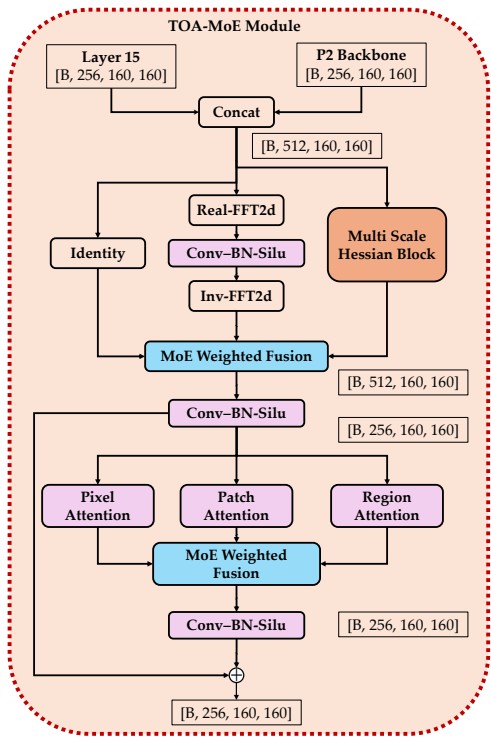

Figure 3: The Proposed TOA-MoE module: Comprises of two stages- 1) **Multi-Domain Feature Enrichment**, processes P2-level features via three parallel experts whose outputs are fused using a **MoE Weighted Fusion**; and 2) **Multi-Granularity Hierarchical Attention**, applies Pixel, Patch, and Region Attention streams, followed by another **MoE Fusion**. A final **residual connection** adds the result back to preserve context and support stable training.

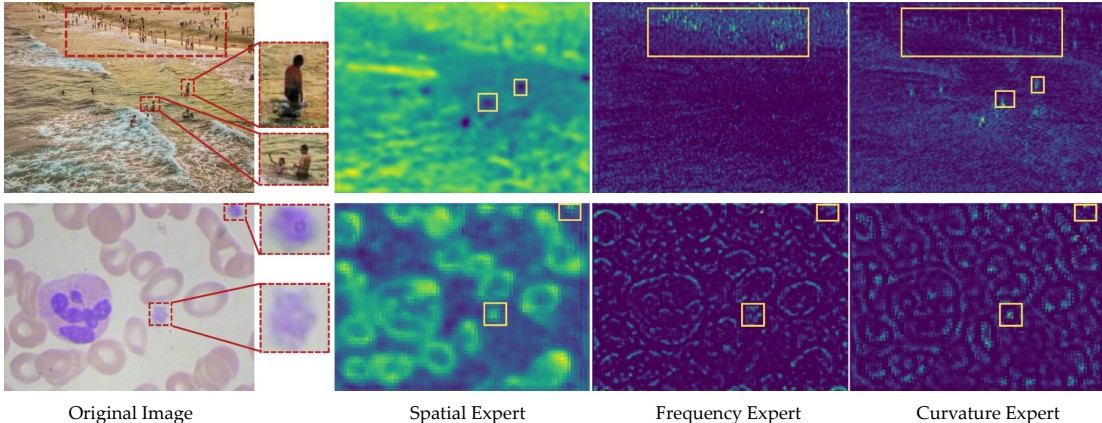

Figure 4: Visualization illustrating the complementary specialization of our Multi-Domain Enrichment experts. On the SeaPerson (top row) and BCCD datasets (bottom row), the Curvature Expert (using Hessian analysis) excels at detecting isolated, blob-like structures (e.g., distant swimmers, platelets), while the Frequency Expert (using FFT) captures repeating textures (e.g., crowds, dense blood cells). This validates our core idea: curvature for particle detection and frequency for pattern detection, together enriching tiny object representation.

a classic mathematical invariant for detecting convex, blob-like regions. The resulting features are concatenated and passed through a $1 \times 1$ convolution and an Squeeze-and-Excitation block Hu et al. (2018) to yield the final output.

To combine these diverse feature representations, we employ our **MoE Weighted Fusion** mechanism. It consists of a lightweight gating network that analyzes the input tensor $X$ to predict the relative importance of each expert. Specifically, the gating branch produces three routing logits $[z_{id}, z_{freq}, z_{hess}] = R(X)$, where $R(\cdot)$ denotes the learnable routing network composed of adaptive pooling and convolutional layers. The logits are normalized using a softmax function to obtain the fusion weights $w_i = \frac{e^{z_i}}{\sum_j e^{z_j}}$, for $i \in \{id, freq, hess\}$.

These weights are then used to compute the enriched feature map:

$$X_{enrich} = w_{id}E_{id}(X) + w_{freq}E_{freq}(X) + w_{hess}E_{hess}(X). \tag{1}$$

### 3.2.2 Stage 2: Hierarchical Attention with MoE Fusion.

The design of our parallel attention streams is conceptually inspired by seminal works like SE-Net Hu et al. (2018) for its channel-wise feature recalibration, and CBAM Woo et al. (2018) for its combination of channel and spatial attention. Our module adapts these principles into a simultaneous, multi-granularity framework. While the enriched feature map, $X_{enrich}$, contains a wealth of multi-domain information, it does not resolve the inherent ambiguity of tiny objects. For instance, distinguishing a clinically relevant platelet from a visually similar speck of imaging noise requires more than just identifying a 'blob'. It demands contextual interpretation at multiple scales. To address this, the second stage of our module employs a hierarchical attention mechanism. The enriched map is passed through a fusion convolution, and the result serves as input ($X_{in}$) to three parallel attention streams, each designed to analyze the features at a distinct semantic granularity from fine boundaries to broad scene context.

- **Pixel-Level Attention:** For an object spanning only a few pixels, its boundary is often the most salient feature distinguishing it from background noise. This stream is engineered to isolate these critical high-frequency details. It utilizes a depth-wise convolution as a learned spatial gradient operator ($\nabla_S$) to sharpen edges and a point-wise MLP ($\phi_{pix}$) to generate local, per-pixel channel weights. The output $X_{pix}$ is given by:

$$X_{pix} = X_{in} \odot \sigma(\phi_{pix}(X_{in} + \nabla_S(X_{in}))) \tag{2}$$

  where $X_{in}$ is the input to this stage, $\sigma$ is the sigmoid function, and $\odot$ is element-wise multiplication. This formulation enhances edge features before selectively amplifying them, improving the model's ability to localize tiny objects precisely.

- **Patch-Level Attention:** While pixels define boundaries, they lack the local context needed to interpret texture or shape. To address this, the patch-level stream analyzes local neighborhoods to learn characteristic patterns, such as the texture of a platelet or the local shape of a cell. It employs a set of parallel depth-wise convolutions with varying kernel sizes, defined by the set $\mathcal{K} = \{k_1, k_2, k_3\}$, to create a scale-robust patch context, $X_{patch}$. An importance weight, $W_{patch}$, is then learned to adaptively interpolate between the original fine-grained features and this new contextual representation:

$$X_{patch} = \frac{1}{|\mathcal{K}|} \sum_{k \in \mathcal{K}} \text{DWConv}_k(X_{in}) \tag{3}$$

$$W_{patch} = \sigma(\phi_{patch}(\text{GAP}(X_{patch}))) \tag{4}$$

$$X_{patch} = (X_{in} \odot W_{patch}) + (X_{patch} \odot (1 - W_{patch})) \tag{5}$$

  where GAP denotes Global Average Pooling. This allows the model to dynamically decide whether to prioritize the original high-frequency features or leverage the new contextual ones.

- **Region-Level Attention:** To resolve visual ambiguities, such as distinguishing a distant person in aerial surveillance from similarly-sized background noise, the model must leverage global scene context. Our region stream addresses this through a novel dual-pathway attention mechanism inspired by CBAM. It simultaneously computes channel attention ($W_{channel}$) from global descriptors

and spatial attention ($W_{spatial}$) using a 7×7 kernel for wide regional receptive fields, then fuses both into a unified attention mask that encodes both *what* features are important and *where* they are located:

$$X' = X_{in} \odot W_{channel} \tag{6}$$

$$X_{region} = X' \odot W_{spatial} \tag{7}$$

To adaptively fuse the pixel, patch, and region-level streams, we apply a **MoE Weighted Fusion** via a lightweight gating network. This module processes the input $X_{\text{in}}$ using global pooling and convolutions to produce softmax normalized weights $\{w_{\text{pix}}, w_{\text{patch}}, w_{\text{region}}\}$. These weights modulate each attention stream after a 1×1 convolution, and the fused output is added residually to $X_{\text{in}}$:

$$X_{\text{weighted}} = \sum_{g \in \{\text{pix, patch, region}\}} w_g \cdot \text{Conv}_g(X_g)$$
$$X_{\text{MOE}} = X_{\text{in}} + \text{Conv}(X_{\text{weighted}}) \tag{8}$$

## 4 Results and Experiments

### 4.1 Experimental Settings

#### 4.1.1 Datasets

This study uses five publicly available datasets to evaluate the performance and robustness of the proposed model: three generic tiny object detection datasets (SeaPerson Yu et al. (2022), TinyPerson Yu et al. (2020) and VisDrone Zhu et al. (2021) datasets), and two blood cell imaging datasets (BCCD Shenggan (2019) and CBC Alam & Islam (2019)) for medical applications of our approach. The official dataset splits were used to ensure fair comparison with prior works and are summarized in Table 1. Additional dataset details are provided in Appendix A.

Table 1: Split distribution of Datasets used in this study

| Dataset | Train | Validation | Test | Total |
|---|---|---|---|---|
| SeaPerson | 5711 | 568 | 5753 | 12032 |
| TinyPerson | 635 | 159 | 816 | 1610 |
| VisDrone | 6471 | 548 | 1610 | 8629 |
| BCCD | 205 | 87 | 72 | 364 |
| CBC | 300 | 60 | 60 | 420 |

#### 4.1.2 Evaluation Metrics

For generic tiny object datasets (SeaPerson, TinyPerson and VisDrone), we evaluate mAP@0.5 along with precision and recall to ensure fair comparison with existing literature and established benchmarks. The precision and recall values are calculated at the standard IoU threshold of 0.5, representing the model's ability to correctly identify objects with at least 50% overlap with ground truth bounding boxes. We use mAP@0.5:0.95 as our primary evaluation metric in blood cell detection task to ensure stricter metric for medical imaging applications because mAP@0.5 can overestimate performance on critical medical tasks by accepting a low IoU threshold (IoU $\geq$ 0.5) as a correct detection, which may include loosely aligned or oversized bounding boxes. mAP@0.5:0.95 averages performance over a range of IoU thresholds (0.5 to 0.95 in steps of 0.05) and thus imposes stricter localization requirements and better captures the model's true detection quality for clinically-sensitive medical tasks. Inference speeds (FPS) were also measured using the TorchScript runtime environment on a single NVIDIA A6000 GPU to measure the model's capability to do real-time inferencing.

#### 4.1.3 Implementation

All experiments were conducted on a system equipped with an Intel Xeon Silver 4410Y CPU (12 cores, 24 threads, 30 MB cache), an NVIDIA RTX A6000 GPU (48 GB VRAM) and 512GB RAM. The software

stack included Python 3.12.2, PyTorch 2.2.2+cu121, and Ultralytics 8.3.63. All models were trained on the SeaPerson dataset for 100 epochs using a batch size of 16, the AdamW optimizer, and a fixed random seed of 42. More details regarding the training hyperparameters for each dataset can be found in Appendix (Section B). Training graph comparison for TOAM-YOLO with YOLOv12 families are illustrated in Figures 5 and 6 (Appendix Section C).

### 4.1.4 Baseline

Since we integrated the proposed TOAM module into the YOLOv12n Tian et al. (2025) architecture, we used YOLOv12 family as a baseline for all datasets to quantify the performance gains by our method. Additionally, we test the changes on YOLOv11nJocher & Qiu (2024) model architecture by incorporating our proposed changes in YOLOv11n architecture. We use different baseline models depending on the established SOTA for each dataset as follows: 1) TPS-YOLO Yao et al. (2025) for the SeaPerson dataset, 2) CST-YOLO Kang et al. (2024) for the TinyPerson dataset, 3) DDH-YOLO Liu & Yang (2025) for the VisDrone dataset, and 4) CST-YOLO (Kang et al. 2024) for the blood cell datasets (BCCD and CBC). A study by Wang et al. (2025a) also reports strong performance on the BCCD dataset but we excluded it from our comparisons due to the unavailability of publicly released code, which prevents independent verification. Given that YOLOv12s has been shown Tian et al. (2025) to exceed the performance of RT-DETR-R18 Zhao et al. (2024) and RT-DETR-R18v2 Lv et al. (2024), we select YOLOv12s as one of our primary baselines in the comparative study. In addition, to provide a direct comparison against transformer-based detectors, we evaluate RT-DETR-L Zhao et al. (2024) as a representative transformer-based detector reference. We include YOLOv12n, YOLOv12s, and YOLOv12m as reference models to discern the performance gains attributable solely to increased model complexity versus those resulting from our systematic architectural changes.

Among the compared methods, YOLOv12 and YOLOv11 variants, as well as CST-YOLO, were reproduced through complete training from scratch. Configuration used for YOLOv12 and YOLOv11 variants is reported in Appendix B. For CST-YOLO Kang et al. (2024), we reproduced the reported results using the publicly available implementation. In contrast, the results of DDH-YOLO Liu & Yang (2025) and TPS-YOLO Yao et al. (2025) are taken directly from the original papers, as their source code is not publicly available.

## 4.2 Experimental Results

Table 2 presents a comparative evaluation of YOLOv12 baseline models and TOAM-YOLO in terms of parameter count, precision, recall, mAP@0.5, and mAP@0.5:0.95 on the SeaPerson dataset, which is the largest dataset in our benchmark. The baseline YOLOv12n model, with 2.6M parameters, achieves a mAP@0.5 of 56.3% and mAP@0.5:0.95 of 22.7%. YOLOv12s, with 9.1M parameters, achieves a mAP@0.5 of 58.4% and mAP@0.5:0.95 of 24.1%. YOLOv12m, with 20.2M parameters and 67.5 GFLOPs, achieves a mAP@0.5 of 60.6%, suggesting that increased model complexity does not necessarily translate to improved performance in this context. Wilcoxon signed-rank tests, reported in Tables 17,18,19 and 20 (Appendix Section B), confirm that TOAM-YOLO and TOE-YOLO significantly outperform both YOLOv12n and YOLOv12m in precision and recall ($p < 0.05$).

Table 2: Comparison of YOLOv12 based baseline models with our changes incorporated in the YOLOv12 architecture on the SeaPerson dataset. Precision and Recall: calculated at IoU$\geq$0.5

| Model | Params (M) | GFLOPs | Precision | Recall | mAP$_{0.5}$ | mAP$_{0.5:0.95}$ |
|---|---|---|---|---|---|---|
| YOLOv12n | 2.6 | 6.5 | 64.9 | 51.2 | 56.3 | 22.7 |
| YOLOv12s | 9.1 | 19.6 | 65.3 | 53.8 | 58.4 | 24.1 |
| YOLOv12m | 20.2 | 67.5 | 66.3 | 55.6 | 60.6 | 25.4 |
| **TOAM-YOLO** | **3.3** | **31.9** | **75.2** | **61.1** | **67.9** | **28.9** |

### 4.2.1 Ablation Study

In this study, we have proposed two variants of the original YOLOv12n dedicated for tiny object detection. First variant, referred to as Tiny object enhanced-YOLO (TOE-YOLO) consists of only the architectural

changes done in the original YOLOv12n, wherein we replace standard upsampling layers with the CARAFE module, integrate DCNv3 blocks for improved spatial adaptability, and introduce an additional detection head optimized for tiny object feature extraction. Second variant, is our main model, namely, TOAM-YOLO wherein we introduce the TOA-MoE module containing the Multi-Domain experts followed by Multi-granularity experts in the TOE-YOLO. To systematically evaluate the contribution of each component, we performed an extensive ablation study comprising a total of ten configurations. This analysis included a progressive enhancement approach, starting with the YOLOv12n baseline, followed by (A) YOLOv12n + CARAFE + DCNv3 + additional detection head (TOE-YOLO), and (B) Configuration A + TOA-MoE module (TOAM-YOLO). Additionally, we conducted eight additional ablations focusing on a detailed constituent analysis of the TOA-MoE module's experts (Frequency and Curvature) and its multi-granularity attention levels (Pixel, Patch, Region), as well as the impact of DCNv3.

Table 3 illustrates the contribution in performance gain provided by the components of the proposed model. Ablations in Table 3 use the hyperparameters listed in Table 15 (Appendix B). From Table 3, we observe that TOE-YOLO (w/o TOA-MoE) introduces the proposed architectural enhancements with a minimal increase in parameters (2.9M), achieving 65.2% mAP@0.5 and 26.7% mAP@0.5:0.95. The full TOAM-YOLO model, which adds TOA-MoE on top of this enhanced TOE-YOLO architecture, achieves the highest overall performance with 75.2% precision, 61.1% recall, 67.9% mAP@0.5, and 28.9% mAP@0.5:0.95 while maintaining a real-time inference speed of 62.36 FPS.

Table 3: Ablation study of the core components in our proposed TOAM-YOLO.

| Model | Params (M) | GFLOPs | Precision | Recall | $mAP_{0.5}$ | $mAP_{0.5:0.95}$ |
|---|---|---|---|---|---|---|
| **TOAM-YOLO (Full Model)** | **3.3** | **31.9** | **75.2** | **61.1** | **67.9** | **28.9** |
| w/o TOA-MoE (TOE-YOLO) | 2.9 | 13.1 | 71.9 | 59.6 | 65.2 | 26.7 |
| w/o DCNv3 | 3.25 | 31.8 | 74.64 | 61.06 | 67.3 | 28.2 |
| w/o Fourier | 3.26 | 30.9 | 74.4 | 60.6 | 66.8 | 28.04 |
| w/o Hessian | 3.1 | 21.3 | 73.36 | 59.3 | 65.5 | 27.4 |
| w/o Fourier + Hessian | 3.05 | 20.2 | 73.4 | 58.7 | 64.9 | 27.09 |
| w/o Pixel-level Attn. | 3.29 | 31.4 | 73.5 | 60.3 | 66.4 | 27.8 |
| w/o Patch-level Attn. | 3.27 | 31.3 | 74.9 | 60.4 | 66.8 | 28.09 |
| w/o Region-level Attn. | 3.27 | 31.6 | 74.4 | 61.1 | 67.6 | 28.4 |
| w/o All Attn. Levels | 3.25 | 30.7 | 73.6 | 61.17 | 67.1 | 28.1 |

Table 4: Progressive architectural ablation on SeaPerson dataset

| Configuration | $mAP_{0.5}$ | $mAP_{0.5:0.95}$ | Params (M) | GFLOPs |
|---|---|---|---|---|
| YOLOv12n | 56.43 | 22.41 | 2.51 | 5.8 |
| YOLOv12n + P2 head | 66.03 | 27.35 | 3.07 | 13.1 |
| YOLOv12n + P2 + CARAFE | 66.19 | 27.40 | 2.83 | 13.0 |
| YOLOv12n + P2 + CARAFE + DCNv3 (TOE-YOLO) | 66.79 | 27.65 | 2.9 | 13.1 |
| TOAM-YOLO (TOE-YOLO + TOA-MoE) | **68.18** | **29.16** | 3.3 | 31.9 |
| TOAM-YOLO without CARAFE and DCNv3 | 66.56 | 27.92 | 3.1 | 31.1 |

To further isolate the contribution of the general architectural enhancements from that of the TOA-MoE module, we additionally report a progressive build-up (Table 4)[1]. Introducing the P2 head alone provides the largest single-step improvement over the YOLOv12n baseline (+9.6% mAP@0.5), while CARAFE and DCNv3 provide additional, smaller gains. The subsequent addition of TOA-MoE (TOAM-YOLO) isolates its incremental contribution on top of this enhanced base. We also performed a reverse ablation by removing CARAFE and DCNv3 from the complete model, while retaining the P2 head and TOA-MoE, which reduces

---

[1]All rows in this sweep, including the baseline, share one fixed batch size of 16, so each delta isolates the added module. Absolute values thus differ marginally from Tables 2 and 3

performance from 67.9% to 66.56% mAP@0.5, confirming that these components contribute positively within the full architecture and complement the TOA-MoE module.

Furthermore, as shown in Table 5, TOAM-YOLO surpasses the recently proposed TPS-YOLO Yao et al. (2025), which achieved a mAP@0.5 of 63.5% on the SeaPerson dataset. To assess model performance uncertainty, we applied bootstrap resampling ($n = 10$) to the SeaPerson test set, yielding a mean mAP@0.5 of $67.3 \pm 0.49$ with a 95% confidence interval of [66.58, 68.02], and a mean mAP@0.5:0.95 of $28.35 \pm 0.3$ with a 95% confidence interval of [27.93, 28.87].

Table 5: Comparison of TOAM-YOLO with other SOTA models on SeaPerson dataset.

| Model | Params (M) | mAP@0.5 |
|---|---|---|
| YOLOv8n* | 3.2 | 50.3 |
| YOLOv8s* | 11.2 | 54.2 |
| YOLOv8m* | 25.9 | 56.0 |
| YOLOv9C* | 25.3 | 54.4 |
| RT-DETR* | 32.8 | 30.7 |
| TPS-YOLO* | 1.1 | 63.5 |
| YOLOv12n | 2.6 | 56.3 |
| YOLOv12s | 9.1 | 58.4 |
| YOLOv12m | 20.2 | 60.6 |
| TOAM-YOLO (Ours) | 3.3 | **67.9** |

[*] Results cited from the TPS-YOLO paper Yao et al. (2025).

To evaluate the generalizability of the proposed modules beyond YOLOv12, we applied the same TOAM enhancements to the YOLOv11 architecture. Table 6 presents the comparative evaluation of YOLOv11 baseline models and TOAM-YOLO on the SeaPerson dataset. TOAM-YOLO built on YOLOv11n achieves 69.52% mAP@0.5 and 29.76% mAP@0.5:0.95, outperforming both YOLOv11n (58.99%) and YOLOv11s (60.53%) by substantial margins. With 3.34M parameters, the YOLOv11-based variant surpasses YOLOv11s (9.43M parameters) by 8.99 percentage points in mAP@0.5 while using 64.6% fewer parameters. This confirms that the proposed architectural changes are backbone-agnostic and yield consistent improvements regardless of the underlying YOLO version.

Table 6: Comparison of YOLOv11 based baseline models with our changes incorporated in the YOLOv11 architecture on the SeaPerson dataset. Precision and Recall: calculated at IoU≥0.5

| Model | Params (M) | GFLOPs | Precision | Recall | $mAP_{0.5}$ | $mAP_{0.5:0.95}$ |
|---|---|---|---|---|---|---|
| YOLOv11n | 2.59 | 6.4 | 65.7 | 53.8 | 58.99 | 24.36 |
| YOLOv11s | 9.43 | 21.5 | 66.54 | 54.41 | 60.53 | 25.7 |
| **TOAM-YOLO (YOLOv11 based)** | **3.34** | **32.4** | **76.2** | **62.79** | **69.52** | **29.76** |

Recent transformer-based detectors have demonstrated strong performance on small-object and dense-scene detection tasks, often at the cost of substantially increased model complexity. To provide a direct comparison under a unified training protocol, RT-DETR-L was selected as a representative transformer-based detector and trained using the Ultralytics framework for 200 epochs on SeaPerson dataset. The results are summarized in Table 7..

TOAM-YOLO (v11n) achieves competitive performance relative to RT-DETR-L, obtaining 69.53% mAP@0.5 compared to 70.87% for RT-DETR-L, while achieving a slightly higher mAP@0.5:0.95 of 29.76% versus 29.50%. Importantly, this performance is obtained with only 3.34M parameters and 32.4 GFLOPs, compared with 32.81M parameters and 108 GFLOPs for RT-DETR-L. Similarly, TOAM-YOLO (v12n) substantially narrows the performance gap relative to its YOLOv12n baseline while maintaining a compact model size. These results suggest that the proposed TOAM modules enable lightweight CNN-based detectors

to achieve accuracy that is competitive with a substantially larger transformer-based model, while requiring approximately 10x fewer parameters and 3.3x lower computational cost. The comparison therefore highlights the favorable accuracy-efficiency trade-off achieved by the proposed framework for tiny-object detection.

Table 7: Comparison with the transformer-based RT-DETR-L detector on the SeaPerson dataset.

| Method | Params (M) | GFLOPs | mAP@0.5 (%) | mAP@0.5:0.95 (%) |
|---|---|---|---|---|
| RT-DETR-L | 32.81 | 108 | 70.87 | 29.50 |
| TOAM-YOLO (YOLOv11n) | 3.34 | 32.4 | 69.53 | 29.76 |
| TOAM-YOLO (YOLOv12n) | 3.30 | 32.0 | 67.90 | 28.92 |

Table 8 demonstrates the performance-efficiency trade-off across the evaluated models with TorchScript runtime on an RTX A6000 GPU. While the YOLOv12 models (n, s, m) exhibit superior FPS, their mAP@0.5 caps at 60.6 (YOLOv12m). In contrast, TOAM-YOLO achieves 67.9% mAP@0.5, a 7.3% improvement over YOLOv12m despite having significantly fewer parameters. This disparity suggests that accuracy is not merely a function of computational scaling, and rather, it underscores the efficacy of our proposed design, which translates computational increase into higher detection precision more effectively than larger architectures like YOLOv12m, all while maintaining a low parameter count. TOAM-YOLO runs at 62.36 FPS (16.04 ms) compared to 89.91 FPS (11.12 ms) for YOLOv12n. These results explicitly quantify the additional computational cost associated with the proposed architecture while demonstrating that the resulting gains in detection accuracy are achieved, while maintaining real-time performance. The difference in inference latency stems from our proposed architecture's current reliance on some operators that are not natively compatible with specialized frameworks like cuDNN or TensorRT. This constraint can be mitigated through low-level optimization of these operations for further speed optimization and substantial latency reduction.

Table 8: Comparison of inference speeds and latency with YOLOv12 variants.

| Model | FPS | Inference Time (ms) | $\mathbf{mAP}_{50}$ |
|---|---|---|---|
| YOLOv12n | 89.91 | 11.12 | 56.3 |
| YOLOv12s | 87.8 | 11.39 | 58.4 |
| YOLOv12m | 76.94 | 13.00 | 60.6 |
| TOE-YOLO | 76.11 | 13.14 | 65.2 |
| TOAM-YOLO | 62.36 | 16.04 | **67.9** |

The mAP@0.5:0.95 scores for RBC, WBC, and platelet detection are summarized in Table 9 across the BCCD and CBC datasets, comparing the performance of CST-YOLO, YOLOv12n, and the proposed TOAM-YOLO model. On the BCCD dataset, TOAM-YOLO achieves the highest overall mAP@0.5:0.95 of 63.5%, outperforming CST-YOLO and YOLOv12n by 2.6% and 1.4%, respectively. TOAM-YOLO achieved mAP of 51.1% on platelets, surpassing CST-YOLO (44.3%) and YOLOv12n (47.3%). It also demonstrates superior performance across all CBC dataset classes, with an overall mAP of 70.6%. It achieves mAP of 74.2% for RBCs, 83.4% for WBCs, and 54.1% for platelets which are 0.9%, 1.3% and 1.7% more than the vanilla YOLOv12n.

Comparison of mAP@0.5 and mAP@0.5:0.95 achieved by TOAM-YOLO and state-of-the-art and baseline models on TinyPerson and Visdrone datasets are represented in Table 10. TOAM-YOLO surpasses the baseline YOLOv12n, YOLOv11n and CST-YOLO models in both mAP@0.5 and mAP@0.5:0.95 achieving 18.24% mAP@0.5 on TinyPerson test set which is 4.64% higher than the baseline YOLOv12n model and 1.34% higher than the much heavier high performing CST-YOLO model. TOAM-YOLO achieved mAP@0.5 of 43.1% on the VisDrone dataset outperforming both the baseline YOLOv12n model (32.1%) and state-of-the-art DDH-YOLO model (40.4%) while also attaining a mAP@0.5:0.95 of 25.43%.

To assess the generalizability of the proposed TOAM modules beyond specialized tiny-object benchmarks, we additionally conducted experiments on the MS-COCO dataset. Both YOLOv12n and TOAM-YOLO were trained with identical hyperparameters for 100 epochs, and evaluated on the val2017 validation set. As reported in Table 11, TOAM-YOLO consistently outperforms YOLOv12n across all COCO metrics. The most pronounced improvements are observed on small objects ($AP_S$: 16.50% $\rightarrow$ 20.40%, $AR_S$: 34.00% $\rightarrow$ 41.30%), followed by moderate gains on medium-sized objects ($AP_M$: +1.8%, $AR_M$: +2.5%), and marginal

Table 9: mAP@0.5:0.95 comparison for blood cell detection in BCCD and CBC datasets.

| Dataset | Model | RBC | WBC | Platelets | Overall |
|---------|-------|-----|-----|-----------|---------|
| BCCD | CST-YOLO | 60.9 | 77.5 | 44.3 | 60.9 |
| | YOLOv12n | **62.5** | 76.6 | 47.3 | 62.1 |
| | TOAM-YOLO | 61.7 | **77.7** | **51.1** | **63.5** |
| CBC | CST-YOLO | 71.3 | 78.6 | 48.6 | 66.2 |
| | YOLOv12n | 73.3 | 82.1 | 52.4 | 69.3 |
| | TOAM-YOLO | **74.2** | **83.4** | **54.1** | **70.6** |

Table 10: Performance comparison of TOAM-YOLO with baselines on TinyPerson and VisDrone dataset.

| Dataset | Model | Params | mAP@0.5 | mAP@0.5:0.95 |
|---------|-------|--------|---------|--------------|
| TinyPerson | CST-YOLO | 48.2M | 16.9 | 5.15 |
| | YOLOv11n | 2.59M | 13.12 | 4.57 |
| | YOLOv12n | 2.55M | 13.6 | 4.79 |
| | TOAM-YOLO | 3.3M | **18.24** | **6.42** |
| VisDrone | DDH-YOLO | 1.67M | 40.4 | 24.2 |
| | YOLOv12n | 2.55M | 32.1 | 18.5 |
| | TOAM-YOLO | 3.3M | **43.1** | **25.43** |

but consistent gains on large objects ($AP_L$: +0.4%, $AR_L$: +0.4%). These results indicate that the proposed enhancement strategy is not restricted solely to highly specialized tiny-object datasets, and that the performance gains remain most pronounced for small and tiny objects, consistent with the original design motivation of TOAM.

Table 11: Generalization evaluation of TOAM-YOLO and YOLOv12n on the MS-COCO val2017 benchmark trained for 100 epochs.

| Method | $AP_{50}$ | $AP_{50:95}$ | $AP_{75}$ | $AP_S$ | $AP_M$ | $AP_L$ | $AR_S$ | $AR_M$ | $AR_L$ |
|--------|-----------|--------------|-----------|--------|--------|--------|--------|--------|--------|
| YOLOv12n | 48.90 | 34.20 | 36.80 | 16.50 | 37.60 | 50.40 | 34.00 | 63.80 | 76.70 |
| TOAM-YOLO (v12n) | **52.40** | **37.50** | **40.90** | **20.40** | **39.40** | **50.80** | **41.30** | **66.30** | **77.10** |

#### 4.2.2 Quantitative Analysis of Expert Specialization

To complement the qualitative visualization in Figure 4, we quantitatively examine the routing behavior of the experts within TOA-MoE. For each evaluated sample, the gating network produces expert-selection weights, and we report the mean, standard deviation, minimum, and maximum routing weights computed over the evaluation split of each dataset (Table 12).

Table 12: Routing-weight statistics (%) per expert across datasets, computed over each evaluation split.

| Dataset | Expert | Mean | Std | Min | Max |
|---------|--------|------|-----|-----|-----|
| SeaPerson | Spatial | 20.98 | 5.88 | 18.55 | 51.83 |
| | Fourier | 8.77 | 2.39 | 4.44 | 19.75 |
| | Hessian | 70.26 | 7.80 | 28.42 | 76.40 |
| VisDrone | Spatial | 8.26 | 0.59 | 7.36 | 11.16 |
| | Fourier | 11.19 | 1.44 | 7.69 | 14.50 |
| | Hessian | 80.55 | 1.72 | 74.34 | 83.86 |
| BCCD | Spatial | 13.73 | 6.71 | 0.53 | 19.28 |
| | Fourier | 3.74 | 2.09 | 0.14 | 5.40 |
| | Hessian | 82.54 | 8.77 | 75.67 | 99.32 |

From Table 12, we note that the Hessian expert consistently receives the largest average routing weight across all datasets. This observation suggests that second-order structural cues, such as local intensity variations, edge responses, and blob-like patterns, provide a broadly useful representation for object localization across

diverse domains. The prominence of the Hessian branch is also consistent with the strong emphasis on boundary and shape information required for detecting small and visually challenging objects.

The distribution of the remaining routing weights, however, exhibits dataset-dependent variation. On SeaPerson and BCCD, the Spatial expert receives a larger average weight than the Fourier expert. Both datasets contain relatively homogeneous object categories and are characterized by spatially localized discriminative features, such as human silhouettes in maritime scenes and cellular morphology in blood smear images. In such settings, preserving local spatial structure appears to be more informative than emphasizing frequency-domain representations.

In contrast, on VisDrone, the Fourier expert becomes the second most utilized branch. VisDrone contains highly diverse urban scenes with substantial scale variation, background clutter, viewpoint changes, and densely packed small objects. Under these conditions, frequency-domain representations may provide complementary information by capturing texture and fine-grained structural patterns that are less apparent in purely spatial features. The increased routing weight assigned to the Fourier expert therefore suggests that the gating mechanism adapts to the greater visual complexity of the dataset.

The statistics also indicate that the model does not collapse to a single expert. On SeaPerson, for example, the Hessian expert's routing weight ranges from 28.42% to 76.40%, while the Spatial expert can receive up to 51.83% of the routing weight for certain samples. These variations indicate that the gating network dynamically balances structural, spatial, and frequency-domain representations according to the visual characteristics of the input image.

These routing statistics are also consistent with the component ablations reported in Table 3. Although the Fourier expert receives a comparatively small average routing weight, removing it still reduces performance (66.8% mAP@0.5 versus 67.9% for the full model), while removing the Hessian expert produces a larger drop (65.5% mAP@0.5). Removing both experts together further decreases performance (64.9% mAP@0.5). The routing-weight statistics and ablation results are therefore mutually consistent: the Hessian expert contributes the largest overall impact, while the Fourier and Spatial experts provide additional information that improves detection performance despite receiving lower average routing weights.

Overall, these findings support the intended specialization behavior of the TOA-MoE design and indicate that expert contributions are adaptively combined rather than converging to a fixed expert preference, consistent with the proposed mixture-of-experts design.

## 5 Discussion

The experimental results demonstrate that TOAM-YOLO improves performance on existing model architectures in tiny object detection across multiple domains. The 11.6% improvement in mAP@0.5 over the baseline YOLOv12n on the SeaPerson dataset represents a significant advancement, particularly considering the challenging nature of detecting small objects in maritime surveillance scenarios where scale variation and environmental conditions pose substantial difficulties. As the progressive ablation (Table 4) shows, this gain is driven primarily by the high-resolution P2 pathway and its supporting architectural changes, with TOA-MoE contributing a further improvement on top of this enhanced TOE-YOLO base. The performance of YOLOv12m suggests that tiny object detection benefits more from architectural innovations than from parameter scaling or increasing the complexity. Our ablation study reveals the complementary nature of the proposed components. The TOE-YOLO variant's 8.9% mAP@0.5 improvement over baseline with minimal parameter overhead (11.6% increase) demonstrates the utility of carefully placed architectural changes on performance. Additionally, the superior performance on platelet detection (51.1% mAP vs 44.3% for CST-YOLO) demonstrates the model's capability to handle extremely small biological structures.

A key finding of this work is that the proposed TOAM enhancements are not specific to the YOLOv12 backbone. When applied to YOLOv11n, TOAM-YOLO achieves 69.52% mAP@0.5 and 29.76% mAP@0.5:0.95 on SeaPerson, surpassing even the YOLOv12-based TOAM-YOLO variant (67.9% mAP@0.5). This result is obtained with a comparable parameter count of 3.34M and confirms that the TOA-MoE module, CARAFE upsampling, DCNv3 integration, and BiFPN-style fusion function as effective plug-and-play components that transfer across YOLO architectures. Thus, both the YOLOv11- and YOLOv12-based TOAM-YOLO variants consistently outperform their respective baselines, demonstrating that the proposed TOAM components

serve as effective plug-and-play enhancements across different detector architectures. The stronger performance of the YOLOv11-based variant may reflect architectural differences between YOLOv11 and YOLOv12 that influence how the proposed TOAM head modules interact with intermediate feature representations. One possible explanation is that the C3k2 and C2PSA components in YOLOv11 provide feature characteristics that are particularly complementary to the TOAM design relative to the A2C2F-based mechanism employed in YOLOv12n. However, the current experiments were not designed to isolate the contribution of individual architectural components, and therefore the precise source of the observed difference cannot be determined conclusively.

The improvements extend beyond maritime surveillance. On the TinyPerson dataset, TOAM-YOLO reaches 18.24% mAP@0.5, outperforming both YOLOv12n (13.6%) and the much larger CST-YOLO (16.9%, 48.2M parameters) while using only 3.3M parameters. On VisDrone, TOAM-YOLO achieves 43.1% mAP@0.5 compared to 32.1% for YOLOv12n and 40.4% for DDH-YOLO, representing an 11.0 percentage point gain over the baseline. These consistent gains across aerial, maritime, and medical domains indicate that changes proposed in TOAM-YOLO can be used to address fundamental limitations in tiny object feature representation instead of simply increasing computational cost by stacking multiple feature extraction layers.

Compared to transformers based approaches, TOAM-YOLO remains parameter efficient while not increasing computational cost drastically with respect to the performance gains. TOAM-YOLO achieves SOTA results among CNN based models on five datasets, with just 3.3M trainable parameters, achieving performance competitive with much larger transformer-based models. Future work will explore increasing the inference speed of the model while maintaining its performance. Further to this, although the present study focuses on closed-set tiny-object detection, an interesting direction for future research is to investigate whether the proposed TOA-MoE framework can be integrated with open-vocabulary detection architectures that leverage vision-language pretraining.

## 6 Conclusion

In this work, we take a step forward in solving tiny object detection challenges across diverse domains. We introduce TOAM-YOLO, a novel framework that increases performance of existing YOLO architectures across five challenging datasets, from aerial surveillance and maritime scenes to medical microscopy. We introduce a novel TOA-MoE module which integrates Multi-domain experts and multi-granularity experts to the YOLOv12n architecture alongside other architectural changes. The efficacy of these contributions is demonstrated through extensive experimentation. On the large-scale SeaPerson benchmark, TOAM-YOLO achieves a mAP@0.5 of 67.9%, with the incorporated P2 pathway accounting for most of this gain and TOA-MoE providing the additional performance improvement, outperforming both the lightweight YOLOv12n baseline and SOTA model. This superior performance is consistent across diverse benchmarks, including TinyPerson, VisDrone, and specialized medical datasets (BCCD and CBC). Our model shows a significant improvement in detecting small objects across different contexts, demonstrating the effectiveness of our proposed enhancements. Collectively, TOAM-YOLO establishes a new benchmark for accurate tiny object detection, proving that a synergistic combination of specialized modules can overcome the limitations of simply scaling model size.

## Broader Impact Statement

TOAM-YOLO is designed to improve the detection of tiny objects in challenging visual environments. Such capability has the potential to benefit a range of applications, including maritime safety, search-and-rescue operations, environmental monitoring, and medical image analysis, where accurate detection of small targets is critical. Several publicly available research benchmark datasets used in this study (e.g., VisDrone, SeaPerson, and TinyPerson) consist of aerial or maritime imagery acquired using drone-based platforms. Technologies developed for these settings may also be deployed for surveillance and monitoring purposes. We encourage practitioners who deploy similar systems to follow applicable privacy regulations, data-protection practices, and institutional oversight, and to restrict use to contexts with an appropriate legal basis and consent. For the medical blood cell detection experiments (BCCD and CBC), we emphasize that the model is a research prototype and has not been clinically validated. It is not intended for diagnostic use and should

not be used as a standalone diagnostic system; any clinical application would require dedicated validation, regulatory approval, and supervision by qualified medical professionals.

## Acknowledgements

The authors gratefully acknowledge the financial support from RailTel Corporation of India Ltd. (Grant Number SRP-286), and the Infosys Centre for Artificial Intelligence, IIIT-Delhi.

## Code Availability

The source code for TOAM-YOLO is publicly available at the following link: https://github.com/Vision-SBILab/TOAM-YOLO.git

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

# A    Dataset Details

This appendix provides additional details and links to the datasets used in this study. The following are the details of the datasets used in this study:

## A.1    SeaPerson Dataset

SeaPerson Yu et al. (2022) is a benchmark dataset for tiny object detection, captured using a drone-mounted camera in seaside environments. It comprises 12,032 high-resolution images (mostly 1920×1080 pixels) and 619,627 annotated person instances. We use the dataset split as described by Yu et al. (2022), which contains 5,711 training images, 568 validation images, and 5,753 testing images.

Dataset link: `https://cove.thecvf.com/datasets/695`

## A.2    TinyPerson Dataset

TinyPerson Yu et al. (2020) is a benchmark tiny object detection dataset containing 1,610 images annotated with high-density human instances, many of which are smaller than 20 pixels. This dataset consists of 794 training images and 816 testing images. For the purpose of training, 80% and 20% of the training images were taken as training and validation sets respectively. Therefore, the final split used was 635 train, 159 validation, and 816 test images.

Dataset link: `http://vision.ucas.ac.cn/sources`

## A.3    VisDrone Dataset

VisDrone is a large-scale benchmark for drone-based image and video analysis tasks, focusing on detection, tracking, and crowd counting in challenging scenarios Zhu et al. (2021). It includes over 2.6 million annotated bounding boxes for pedestrians, vehicles, and other urban objects. For the detection task, the dataset is split into 6,471 training images, 548 validation images, and 1,610 test-development images.

Dataset link: `https://github.com/VisDrone/VisDrone-Dataset`

## A.4    BCCD Dataset

The BCCD dataset Shenggan (2019) is a small-scale blood cell detection dataset consisting of 364 images of resolution 640×480. The split includes 205 training images, 87 validation images, and 72 testing images.

Dataset link: `https://github.com/Shenggan/BCCD_Dataset`

## A.5    CBC Dataset

The CBC dataset Alam & Islam (2019) contains 300 training images, 60 validation images, and 60 testing images, all of resolution 640×480. These blood cell datasets assess model effectiveness in biomedical image analysis, especially platelet detection, and demonstrate the framework's versatility in biomedical image analysis.

Dataset link:    `https://github.com/MahmudulAlam/Complete-Blood-Cell-Count-Dataset?tab=readme-ov-file`

# B    Technical Details

## B.1    Hyperparameters

Table 13: Training hyperparameters used for training YOLOv11n, YOLOv11s, YOLOv12s, YOLOv12m and TOAM-YOLO models on SeaPerson dataset

| Hyperparameter | Value |
|---|---|
| Image Size | $640 \times 640$ |
| Epochs | 100 |
| Batch Size | 16 |
| Optimizer | AdamW |
| Initial Learning Rate ($lr_0$) | 0.01 |
| Final Learning Rate ($lrf$) | 0.01 |
| Cos_lr | False |
| AMP | Disabled |
| Patience | 50 |
| Seed | 42 |
| Validation/Test Batch Size | 8 |
| Augmentations | Default |

Table 14: Training hyperparameters used for training YOLOv12n model on SeaPerson dataset

| Hyperparameter | Value |
|---|---|
| Image Size | $640 \times 640$ |
| Epochs | 100 |
| Batch Size | 80 |
| Optimizer | AdamW |
| Initial Learning Rate ($lr_0$) | 0.01 |
| Final Learning Rate ($lrf$) | 0.01 |
| Cos_lr | False |
| AMP | Disabled |
| Patience | 50 |
| Seed | 42 |
| Validation/Test Batch Size | 8 |
| Augmentations | Default |

Table 15: Training hyperparameters used for training YOLOv12n and TOAM-YOLO Models on TinyPerson, VisDrone, BCCD and CBC datasets

| Hyperparameter | TinyPerson | VisDrone | BCCD | CBC |
|---|---|---|---|---|
| Image Size | $640 \times 640$ | $640 \times 640$ | $640 \times 640$ | $640 \times 640$ |
| Epochs | 500 | 500 | 500 | 500 |
| Batch Size | 24 | 16 | 24 | 24 |
| Optimizer | AdamW | AdamW | AdamW | AdamW |
| Initial Learning Rate ($lr\_0$) | 0.01 | 0.01 | 0.01 | 0.01 |
| Final Learning Rate ($lr\_f$) | 0.01 | 0.01 | 0.01 | 0.01 |
| Cos_lr | False | False | True | False |
| AMP | Disabled | Disabled | Disabled | Disabled |
| Patience | 50 | 50 | 50 | 50 |
| Seed | 42 | 42 | 42 | 42 |
| Validation/Test Batch Size | 8 | 8 | 8 | 8 |
| Augmentations | Default | Default | Custom | Custom |

*Custom augmentation parameters (BCCD & CBC):* `hsv_h`=0.015, `hsv_s`=0.2, `hsv_v`=0.7, `mosaic`=0.4, `erasing`=0.4, `shear`=0.0, `scale`=0.5, `close_mosaic`=15

Table 16: CST-YOLO training hyperparameters used for TinyPerson and BCCD datasets. *(Default)*

| Hyperparameter | TinyPerson | BCCD |
|---|---|---|
| Image Size | $640 \times 640$ | $640 \times 640$ |
| Epochs | 500 | 500 |
| Batch Size | 24 | 24 |
| Optimizer | SGD | SGD |
| Initial Learning Rate ($lr_0$) | 0.01 | 0.01 |
| Final Learning Rate ($lr_f$) | 0.1 | 0.1 |
| Momentum | 0.937 | 0.937 |
| Weight Decay | 0.0005 | 0.0005 |
| Warmup Epochs | 3.0 | 3.0 |
| Mosaic | 1.0 | 1.0 |
| Mixup | 0.15 | 0.15 |
| Paste-in | 0.15 | 0.15 |
| *Additional augmentations (default):* `hsv_h` $= 0.015$, `hsv_s` $= 0.7$, `hsv_v` $= 0.4$, `translate` $= 0.2$, `scale` $= 0.9$, `fliplr` $= 0.5$, `copy_paste` $= 0.0$ | | |

**RT-DETR-L Training Configuration**  RT-DETR-L was trained from scratch on the SeaPerson dataset using the Ultralytics framework for 200 epochs with a batch size of 16. All other training settings followed the default RT-DETR-L configuration provided by the framework.

### B.2 Wilcoxon Signed-Rank Test

To assess the statistical significance of per-image performance differences between our proposed method and the baselines, we employed the Wilcoxon signed-rank test Wilcoxon (1992), a non-parametric test suitable for matched-pair data without assuming normality.

Let $\{(x_i, y_i)\}_{i=1}^n$ denote $n$ pairs of observations, and define the paired differences as $d_i = x_i - y_i$.

**Hypotheses.**

- Null hypothesis ($H_0$): The distribution of $d_i$ is symmetric about zero.

- Alternative hypothesis ($H_1$): The distribution of $d_i$ is not symmetric about zero.

**Test Procedure.**  Assuming $d_i \neq 0$ for all $i$ (i.e., zero-differences are excluded), the test is conducted as follows:

1. Compute the differences: $d_i = x_i - y_i$.

2. Compute the absolute differences: $|d_i|$.

3. Rank the absolute differences in ascending order (ties receive average ranks).

4. Assign the original signs of $d_i$ to the corresponding ranks.

5. Compute the test statistic:
$$W = \min(W^+, W^-)$$
where $W^+$ is the sum of the ranks for $d_i > 0$ and $W^-$ is the sum of the ranks for $d_i < 0$.

**P-value Computation.**  The p-value is computed using:

- The exact distribution of $W$ for small sample sizes ($n$).

- A normal approximation for larger $n \gtrsim 20$, using:

$$Z = \frac{W - \frac{n(n+1)}{4}}{\sqrt{\frac{n(n+1)(2n+1)}{24}}}$$

On all 4 metrics on SeaPerson dataset, our methods (Both TOAM-YOLO and TOE-YOLO) significantly outperformed the baseline YOLOv12n ($p < 0.05$), confirming the robustness of the observed improvements. The evaluation procedure involved assessing whether two object detection models exhibit statistically significant differences in performance. First, the test set (5753 images) was split into 58 non-overlapping batches of 100 images each. Each batch was evaluated independently by both models, and four performance metrics: Precision, Recall, mAP@0.5, and mAP@0.5:0.95 were recorded for them. These per-batch scores formed paired samples for each metric across all batches. The Wilcoxon signed-rank test was then applied to the paired metric values to determine whether the median of their differences deviates significantly from zero. This test is non-parametric and suited for matched samples without assuming normality. The resulting test statistics and p-values provide a measure of whether one model consistently outperforms the other, and were recorded for analysis and reporting.

Table 17: Wilcoxon signed rank test between YOLOv12n and TOAM-YOLO on SeaPerson dataset.

| Parameter | YOLOv12n | TOAM-YOLO | P-Value |
|---|---|---|---|
| mAP@0.5:0.95 | 22.7 | 28.9 | $6.36 \times 10^{-17}$ |
| mAP@0.5 | 56.3 | 67.9 | $4.20 \times 10^{-17}$ |
| Precision | 64.9 | 75.2 | $2.27 \times 10^{-14}$ |
| Recall | 51.2 | 61.1 | $3.19 \times 10^{-16}$ |

Table 18: Wilcoxon signed rank test between YOLOv12n and TOE-YOLO on SeaPerson dataset.

| Parameter | YOLOv12n | TOE-YOLO | P-Value |
|---|---|---|---|
| mAP0.5:0.95 | 22.7 | 26.7 | $6.13 \times 10^{-16}$ |
| mAP0.5 | 56.3 | 65.2 | $1.17 \times 10^{-16}$ |
| Precision | 64.9 | 71.9 | $8.99 \times 10^{-9}$ |
| Recall | 51.2 | 59.6 | $1.45 \times 10^{-15}$ |

Table 19: Wilcoxon signed rank test between YOLOv12m and TOAM-YOLO on SeaPerson dataset.

| Parameter | YOLOv12m | TOAM-YOLO | P-Value |
|---|---|---|---|
| mAP0.5:0.95 | 25.4 | 28.9 | $3.61 \times 10^{-12}$ |
| mAP0.5 | 60.6 | 67.9 | $1.56 \times 10^{-12}$ |
| Precision | 66.3 | 75.2 | $2.02 \times 10^{-10}$ |
| Recall | 55.6 | 61.1 | $1.54 \times 10^{-12}$ |

Table 20: Wilcoxon signed rank test between YOLOv12m and TOE-YOLO on SeaPerson dataset.

| Parameter | YOLOv12m | TOE-YOLO | P-Value |
|---|---|---|---|
| mAP0.5:0.95 | 25.4 | 26.7 | $5.8 \times 10^{-4}$ |
| mAP0.5 | 60.6 | 65.2 | $9.03 \times 10^{-7}$ |
| Precision | 66.3 | 71.9 | $3.02 \times 10^{-3}$ |
| Recall | 55.6 | 59.6 | $1.96 \times 10^{-8}$ |

## B.3 Bootstrap-Based Confidence Interval Estimation for Detection Metrics

To assess the reliability of detection metrics obtained on the SeaPerson dataset, we conducted a bootstrap analysis Efron & Tibshirani (1994) to estimate the empirical sampling distribution of the mean average precision (mAP). This approach allows us to compute non-parametric confidence intervals around the model's performance without assuming any parametric form of the underlying distribution of scores.

We resampled the evaluation set with replacement for N=1000 bootstrap iterations. In each iteration, mAP@50 and mAP@50:95 were computed for a randomly sampled validation set using the TOAM-YOLO model.

---

**Algorithm 1:** Bootstrap Estimation of Detection Metric Confidence Intervals

---

**Input:** Model $\mathcal{M}$, image set $\mathcal{D}$ of size $n$, iterations $B$
**Output:** Empirical distributions and 95% confidence intervals for detection metrics
Initialize: $\mathcal{L}_{50} \leftarrow [\,]$, $\mathcal{L}_{50:95} \leftarrow [\,]$;
**for** $i \leftarrow 1$ *to* $B$ **do**
    Sample $\mathcal{D}_i \sim \mathcal{D}$ with replacement, $|\mathcal{D}_i| = n$;
    Write sampled image paths to file; update YAML $\mathcal{Y}_i$;
    Evaluate $\mathcal{M}$ on $\mathcal{Y}_i$ to obtain $m_i^{50}, m_i^{50:95}$;
    Append $m_i^{50}$ to list $\mathcal{L}_{50}$ and $m_i^{50:95}$ to list $\mathcal{L}_{50:95}$;
    Delete temporary files;
**end**
Compute mean, std, and percentiles from both $\mathcal{L}_{50}$ and $\mathcal{L}_{50:95}$.
**return** $(\mu, \sigma, \mathrm{CI}_{2.5}, \mathrm{CI}_{97.5})$ for each metric.

---

Table 21: Bootstrap-derived statistics for mAP scores of TOAM-YOLO on the SeaPerson dataset. Results are reported as percentages with confidence intervals computed from the 2.5th and 97.5th percentiles across $N$ bootstrap samples.

| Metric | Dataset | Mean | Std | $\mathbf{CI}_{low}$ | $\mathbf{CI}_{up}$ |
|--------|---------|------|-----|------|------|
| mAP@0.5 | SeaPerson ($N = 10$) | 67.30% | 0.49% | 66.58% | 68.02% |
| | SeaPerson ($N = 1000$) | 67.51% | 0.45% | 66.59% | 68.37% |
| mAP@0.5:0.95 | SeaPerson ($N = 10$) | 28.35% | 0.30% | 27.93% | 28.87% |
| | SeaPerson ($N = 1000$) | 28.44% | 0.25% | 27.96% | 28.92% |

## B.4 Curvature Expert - Multi Scale Hessian Block

---

**Algorithm 2:** Multi-Scale Hessian Feature Extraction

---

**Input:** $X \in \mathbb{R}^{C \times H \times W}$
  **Output:** $F_{\text{out}}$
Scales: $\mathcal{D} = \{1, 2, 3\}$;
Kernels:

$$K_{xx} = \begin{bmatrix} 0 & 0 & 0 \\ 1 & -2 & 1 \\ 0 & 0 & 0 \end{bmatrix}, \quad K_{yy} = \begin{bmatrix} 0 & 1 & 0 \\ 0 & -2 & 0 \\ 0 & 1 & 0 \end{bmatrix}, \quad K_{xy} = \frac{1}{4}\begin{bmatrix} 1 & 0 & -1 \\ 0 & 0 & 0 \\ -1 & 0 & 1 \end{bmatrix}$$

**Initialize:** feature list $\mathcal{F} \leftarrow [\,]$;
**foreach** $d \in \mathcal{D}$ **do**
    $F_{xx} \leftarrow \mathrm{DWConv}(X, K_{xx}, d)$; $F_{yy} \leftarrow \mathrm{DWConv}(X, K_{yy}, d)$; $F_{xy} \leftarrow \mathrm{DWConv}(X, K_{xy}, d)$;
    $DoH \leftarrow F_{xx} \cdot F_{yy} - F_{xy}^2$;
    append $(F_{xx}, F_{yy}, F_{xy}, DoH)$ to $\mathcal{F}$
$F_{\text{all}} \leftarrow \mathrm{Concat}(\mathcal{F})$;
$F_{\text{fused}} \leftarrow \mathrm{Conv1x1}(F_{\text{all}})$;
**return** $F_{\text{out}} \leftarrow \mathrm{SEBlock}(F_{\text{fused}})$;

---

### B.4.1 Theoretical Foundations of Hessian Blob Detection

At the core of Hessian blob detection Marsh et al. (2018) lies the Hessian matrix, which is a square matrix of second-order partial derivatives of a scalar-valued function. For an image $I(x, y)$, viewed as a 2D function,

the Hessian matrix $\mathbf{H}(x, y)$ is defined as:

$$\mathbf{H}(x, y) = \begin{pmatrix} \frac{\partial^2 I}{\partial x^2} & \frac{\partial^2 I}{\partial x \partial y} \\ \frac{\partial^2 I}{\partial y \partial x} & \frac{\partial^2 I}{\partial y^2} \end{pmatrix}$$

For continuous and sufficiently smooth image functions, the mixed partial derivatives are equal, i.e., $\frac{\partial^2 I}{\partial x \partial y} = \frac{\partial^2 I}{\partial y \partial x}$, making the Hessian matrix symmetric.

### B.4.2 Approximation of Second-Order Derivatives: Rationale for Kernel Selection

In digital image processing Xu (2014), these continuous derivatives are approximated using discrete convolution kernels. The **Hessian Stream block** utilizes specific 3x3 kernels rooted in the principles of **finite difference approximations** to estimate the derivatives from discrete pixel data.

- **Second derivative with respect to $x$ ($\frac{\partial^2 I}{\partial x^2}$):** This derivative measures the curvature of the image intensity in the horizontal direction. The kernel represents a **central finite difference approximation**, derived from applying the first-order central difference twice. It approximates the derivative as $I(x + 1) - 2I(x) + I(x - 1)$, which is captured by the `[1, -2, 1]` pattern.

$$\mathbf{K}_{xx} = \begin{pmatrix} 0 & 0 & 0 \\ 1 & -2 & 1 \\ 0 & 0 & 0 \end{pmatrix}$$

- **Second derivative with respect to $y$ ($\frac{\partial^2 I}{\partial y^2}$):** Similarly, this kernel measures the curvature in the vertical direction, applying the same central finite difference logic along the y-axis.

$$\mathbf{K}_{yy} = \begin{pmatrix} 0 & 1 & 0 \\ 0 & -2 & 0 \\ 0 & 1 & 0 \end{pmatrix}$$

- **Mixed second derivative ($\frac{\partial^2 I}{\partial x \partial y}$):** This derivative quantifies how the rate of change in one direction varies along the perpendicular direction. The kernel is derived from a **cross-shaped finite difference approximation**. The scaling factor of 0.25 arises from normalization when approximating the derivative over a local pixel grid.

$$\mathbf{K}_{xy} = 0.25 \times \begin{pmatrix} 1 & 0 & -1 \\ 0 & 0 & 0 \\ -1 & 0 & 1 \end{pmatrix}$$

### B.4.3 Determinant of Hessian (DoH)

The **Determinant of Hessian (DoH)** is a powerful measure for identifying blob-like structures. It is computed from the Hessian matrix elements as:

$$\text{DoH}(x, y) = \det(\mathbf{H}(x, y)) = \left( \frac{\partial^2 I}{\partial x^2} \right) \left( \frac{\partial^2 I}{\partial y^2} \right) - \left( \frac{\partial^2 I}{\partial x \partial y} \right)^2$$

A high absolute value of the DoH indicates a strong change in image intensity, characteristic of a blob. Specifically, a large positive DoH value often corresponds to the center of a blob, because at such points, the principal curvatures (eigenvalues of the Hessian) are either both positive (dark blob on light background) or both negative (bright blob on dark background), leading to a positive product. The DoH is a scale-covariant measure, making it suitable for multi-block analysis of **tiny objects**.

### B.4.4   Multi-Block Hessian Stream in the Object Detection Pipeline

The **Multi-Block Hessian Stream** is a critical component designed to extract robust Hessian features across different receptive fields. This multi-block approach is crucial for **tiny object detection** because small objects can appear with varying characteristics that a single receptive field might miss.

### B.4.5   Blocks and Dilated Convolutions

Within this stream, "blocks" correspond to different **dilation rates** in the convolution operation. Dilation (also known as atrous convolution) expands the receptive field of the kernel without increasing parameters or sacrificing resolution. Using dilation rates of $1, 2, 3$, the $3 \times 3$ kernels operate over effective areas of $3 \times 3$, $5 \times 5$, and $7 \times 7$ pixels, respectively. This allows the same kernels to detect blob-like features at multiple scales. Padding is set equal to the dilation rate to maintain the output feature map size, ensuring spatial alignment.

### B.4.6   Feature Computation and Fusion for Tiny Objects

The process for each block (corresponding to a dilation rate $d$) is as follows:

- The input feature map is convolved with $\mathbf{K}_{xx}$, $\mathbf{K}_{yy}$, and $\mathbf{K}_{xy}$ using the corresponding dilation rate to obtain feature maps $f_{xx}$, $f_{yy}$, and $f_{xy}$.
- The Determinant of Hessian (DoH) is then computed as $f_{xx} \cdot f_{yy} - (f_{xy})^2$.

These four features ($f_{xx}$, $f_{yy}$, $f_{xy}$, and DoH) from all blocks are then concatenated along the channel dimension. A $1 \times 1$ convolution (`fusion_conv`) is applied to fuse these diverse features into a compact and informative representation. Finally, a Squeeze-and-Excitation (SE) block (`attention`) applies channel-wise attention, adaptively re-calibrating feature responses to allow the pipeline to focus on the most relevant features for **tiny object recognition**.

## C   Training Graphs of the Models

Figure 5 and Figure 6 show the trend of mAP@0.5 and mAP@0.5:0.95 respectively during training. It can be observed that TOE-YOLO and TOAM-YOLO both improve in performance over YOLOv12n, YOLOv12s and YOLOv12m.

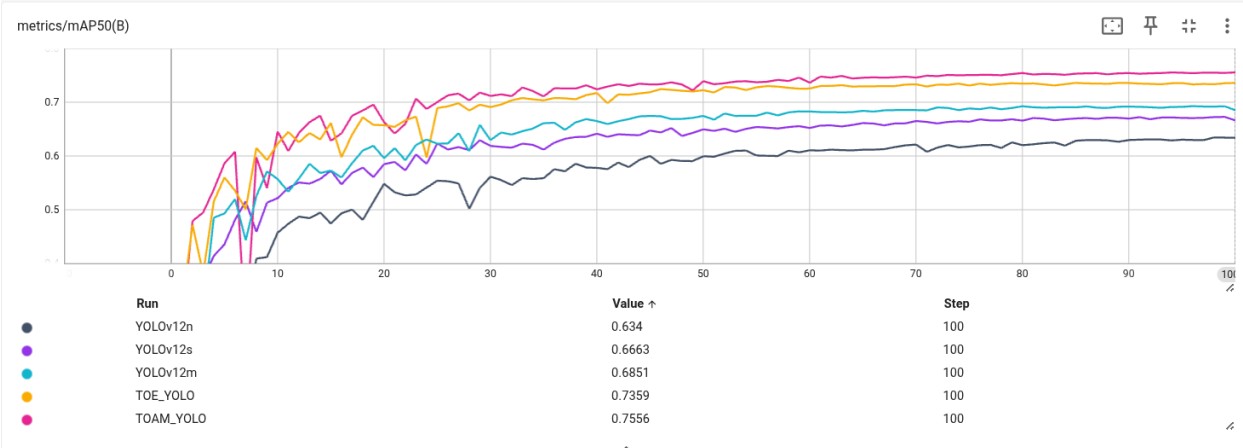

Figure 5: mAP@0.5 trend of YOLOv12n, YOLOv12s, YOLOv12m, TOE-YOLO and TOAM-YOLO during training.

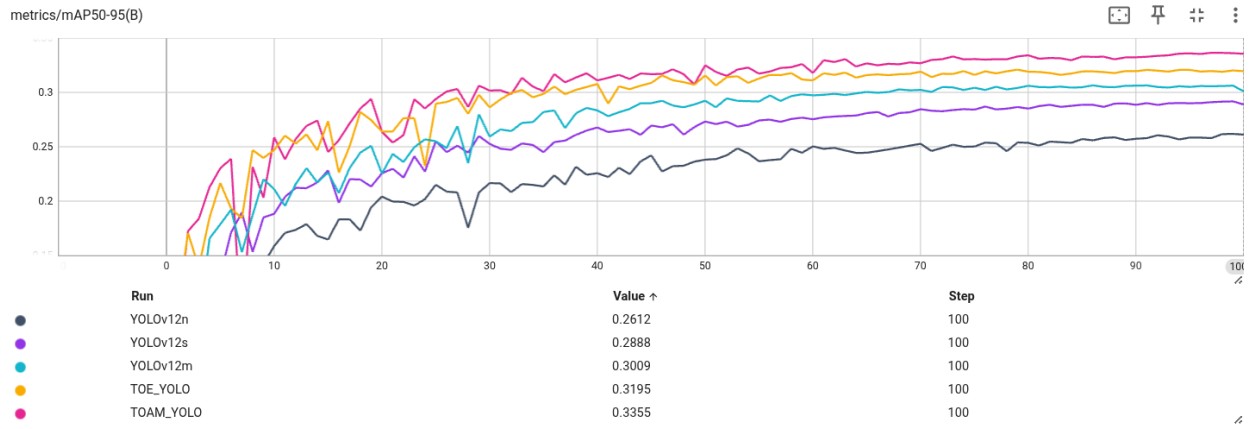

Figure 6: mAP0.5:0.95 trend of YOLOv12n, YOLOv12s, YOLOv12m, TOE-YOLO and TOAM-YOLO during training.

## C.1  Model Prediction Comparisons

Figure 7 presents a qualitative comparison of predictions on two images from the SeaPerson dataset, showing the outputs of YOLOv12n and TOAM-YOLO relative to the Ground Truth.

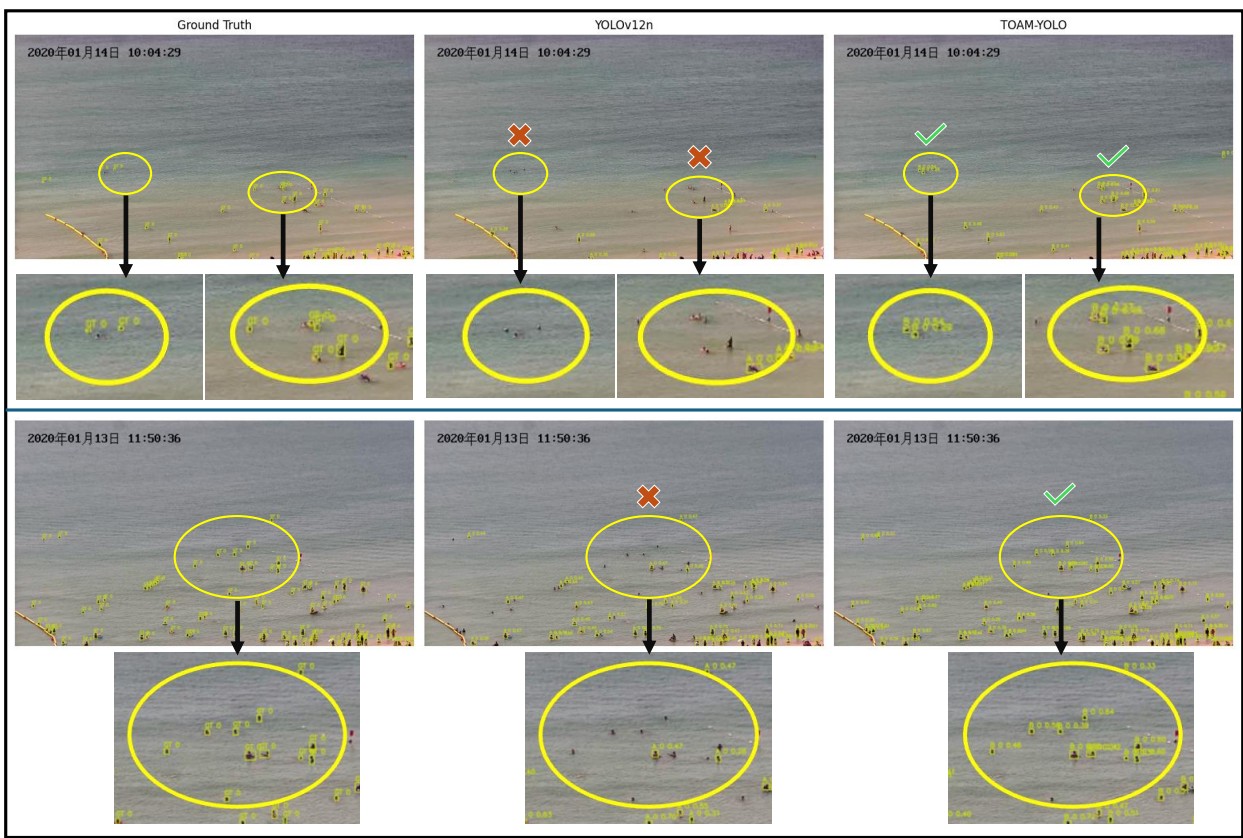

Figure 7: Qualitative comparison of detection results across Ground Truth (Left), YOLOv12n (Center), and TOAM-YOLO (Right). The highlighted regions illustrate that TOAM-YOLO enhances the detection of small and partially occluded objects compared to YOLOv12n. Enlarged crops are provided to facilitate a clearer comparison of both models' predictions relative to the Ground Truth.

