# OpenReview forum: "TOAM-YOLO: A Tiny Object-Aware Multi-Expert YOLO Framework for Diverse Domains"
_TMLR — Accepted by TMLR_

### Review · Reviewer_oges · 2026-05-07

**Summary Of Contributions:**

This paper introduces TOAM-YOLO, a specialized framework designed to address the persistent challenges of tiny object detection across diverse domains, including maritime, aerial, and biomedical imaging. The core innovation is the TOA-MOE (Tiny Object Aware Mixture of Experts) module, which uses a two-stage approach: first enriching features through spatial, frequency (Fourier-based), and curvature (Hessian-based) experts, and then refining them using a 3-level hierarchical attention mechanism. The authors also incorporate a high-resolution P2 detection head, CARAFE upsampling, and a BiFPN-style feature fusion network with DCNv3. The model achieves state-of-the-art performance on five datasets.


Strength:
 - The integration of a curvature expert (Hessian-based) and a frequency expert (Fourier-based) specifically targets the "blob-like" and textural characteristics of tiny objects, which standard CNNs often miss.
 - The paper demonstrates excellent generalizability by achieving top-tier results in drastically different environments, from maritime surveillance to microscopic blood cell analysis.
 - The framework is parameter-efficient and "plug-and-play," showing consistent performance gains when applied to both YOLOv11 and YOLOv12 backbones.


Weakness:
 - The paper acknowledges that some proposed operators are not natively compatible with specialized inference frameworks like TensorRT or cuDNN, resulting in a lower FPS compared to vanilla YOLOv12 models.
 - This work does not provide comparison with large vision language models to show relative performance.
 - For the YOLOv11-based variant, the authors note that further investigation is needed to isolate why it outperforms the YOLOv12 variant despite the latter's more advanced baseline.

**Additional Comments:**

There are not concerns that that would require adding a Broader Impact Statement.

**Audience:**

Yes

**Audience Explanation:**

The researchers that are interested in the detection tasks will be willing to review this paper.

**Broader Impact Concerns:**

There are not concerns that that would require adding a Broader Impact Statement.

**Claims And Evidence:**

Yes

**Claims Explanation:**

The authors provide a thorough experimental evaluation across five distinct datasets and include a detailed ablation study.

**Requested Changes:**

- Add experiments on time complexity comparison among different methods.
 - Improve layout of the paper, current version contains too much space.
 - Provide a simple comparison with current state-of-the-art large vision large language models.

---

> ### Author Response · Authors · 2026-06-01
> **Response to Reviewer oges**
>
> We thank the Reviewer for their constructive feedback and insightful suggestions. We have carefully addressed each point to improve the technical depth and presentation of the manuscript.
>
> RC: Requested Change; W: Weakness
>
> > 1. Add experiments on time complexity comparison among different methods.
>
> RC1. We thank the reviewer for this suggestion. We have added the below Table A1 on inference-time comparison to the revised manuscript (Refer to Table-8 of the revised manuscript). This Table reports the measured FPS and corresponding inference latency for all evaluated YOLOv12-based variants, together with their mAP\@0.5 performance:
>
> **Table A1: Inference speed and latency across models**
> | Model     | FPS   | Inference Time (ms) | mAP50 |
> | --------- | ----- | ------------------- | ----- |
> | YOLOv12n  | 89.91 | 11.12               | 56.3  |
> | YOLOv12s  | 87.8  | 11.39               | 58.4  |
> | YOLOv12m  | 76.94 | 13                  | 60.6  |
> | TOE-YOLO  | 76.11 | 13.14               | 65.2 |
> | TOAM-YOLO | 62.36 | 16.04               | 67.9  |
>
> > 2. Improve layout of the paper, current version contains too much space.
>
> RC2. We thank the reviewer for pointing this out. We have reviewed the manuscript formatting and corrected excessive whitespace around tables and figures, as well as other layout inconsistencies. The revised manuscript has been reformatted to improve readability and overall presentation quality.
>
> > 3. Provide a simple comparison with current state-of-the-art large vision large language models.
>
> RC3 & W2. We thank the reviewer for this suggestion. Large vision-language models and open-vocabulary detectors, such as Grounding DINO and GLIP, represent an important and rapidly evolving direction in visual recognition research. However, the primary objective of this work is closed-set tiny-object detection, where models are trained and evaluated on a predefined set of object categories. In contrast, open-vocabulary vision-language models rely on language supervision and large-scale pretraining to generalize to unseen categories, operating under a fundamentally different training, inference, and evaluation paradigm.
> Consequently, direct performance comparisons would be difficult to interpret, as any observed differences could reflect the underlying learning framework rather than the architectural contributions of the proposed method. To ensure a fair and meaningful evaluation, we therefore compare TOAM-YOLO against established object detection and dedicated tiny-object detection methods that address the same task under the same closed-set setting.
> While a comprehensive comparison with open-vocabulary vision-language models would be of interest, such an investigation would require a different experimental protocol and falls beyond the scope of the current manuscript. We nevertheless acknowledge the growing importance of open-vocabulary detection and consider extending TOAM-YOLO to, or evaluating it within, such frameworks to be an interesting direction for future work. This has been added to the revised manuscript in the last para of the Discussion Section and highlighted in blue.
>
> > 4. For the YOLOv11-based variant, the authors note that further investigation is needed to isolate why it outperforms the YOLOv12 variant despite the latter's more advanced baseline.
>
> W3. We thank the reviewer for raising this important point. As discussed in Section 5 (Paragraph 2, highlighted in blue in the revised manuscript), the stronger performance of the YOLOv11-based variant may be attributable to architectural differences between YOLOv11 and YOLOv12 that influence how the proposed TOAM modules interact with intermediate feature representations. However, the current study was not designed to isolate the effects of individual backbone or neck components of YOLOv11- and YOLOv12-based variants, and therefore the precise cause of the observed difference cannot be determined conclusively. Importantly, both the YOLOv11- and YOLOv12-based TOAM-YOLO variants consistently outperform their respective baselines, demonstrating that the proposed TOAM components serve as effective plug-and-play enhancements across different detector architectures.

---

### Review · Reviewer_u7Cs · 2026-05-09

**Summary Of Contributions:**

This paper proposes TOAM-YOLO, a YOLO-based framework for tiny object detection across maritime, aerial, and medical imaging domains. It builds on YOLOv12n and introduces several architectural enhancements, including a high-resolution P2 detection head, CARAFE upsampling, DCNv3 blocks, BiFPN-style feature fusion, and a TOA-MoE module. The TOA-MoE module combines spatial, Fourier-frequency, and Hessian-curvature experts, followed by pixel-level, patch-level, and region-level attention. The paper evaluates the framework on five datasets: SeaPerson, TinyPerson, VisDrone, BCCD, and CBC. The reported results show notable improvements over YOLOv12n and several task-specific baselines, especially on SeaPerson and VisDrone. The paper also includes useful ablation studies and statistical analyses.

The main **strengths** are the practical relevance of the problem, the broad evaluation across multiple datasets, and the strong empirical gains on several tiny object benchmarks. The main **weaknesses** are that many modules are introduced together, making the individual contribution of each component less clear; some claims about efficiency and SOTA performance require more careful framing; and some numerical descriptions appear inconsistent with the reported tables.

**Additional Comments:**

I appreciate the authors' effort in evaluating the framework across multiple datasets and providing ablations and statistical analyses. My main suggestions are intended to improve claim clarity and contribution attribution.

**Audience:**

Yes

**Audience Explanation:**

Tiny object detection is a practically important problem, and the paper studies it in several relevant domains, including maritime surveillance, aerial imagery, and medical blood cell detection. The findings should be of interest to readers working on small/tiny object detection, efficient object detection, YOLO-based architectures, and applied computer vision systems.

In particular, the paper provides empirical evidence that targeted architectural modifications can improve tiny object detection beyond simply scaling YOLO models. Even if the contribution is mainly an integration of several existing ideas, the cross-dataset evaluation and the reported gains make the results useful for at least part of the TMLR audience.

**Broader Impact Concerns:**

The paper does not raise major ethical concerns. Still, since the method is evaluated on maritime and drone imagery, it may be useful to briefly acknowledge potential privacy or surveillance-related concerns in a broader impact statement. For the medical blood cell detection experiments, the paper should also clarify that the model is not clinically validated and should not be used as a standalone diagnostic system.

**Claims And Evidence:**

Yes

**Claims Explanation:**

The main empirical claim of the paper is that TOAM-YOLO improves tiny object detection performance over YOLOv12n and several existing tiny-object-oriented baselines. This claim is generally supported by the experimental evidence. The paper evaluates the method on five datasets from different domains, including maritime scenes, aerial imagery, and medical blood cell images. The reported improvements are especially clear on SeaPerson and VisDrone, where TOAM-YOLO achieves substantial gains over YOLOv12n and also compares favorably with task-specific baselines. The inclusion of ablation studies, Wilcoxon signed-rank tests, and bootstrap confidence intervals on SeaPerson further strengthens the empirical support.

- My reservations are mainly about the precision of some stronger claims rather than the overall effectiveness of the framework. **Since the method introduces several architectural changes together, the current evidence supports TOAM-YOLO as an effective overall framework, but it is less clear how much of the gain should be attributed specifically to TOA-MoE**. In particular, TOE-YOLO already contributes a large part of the improvement, so a finer decomposition of the general architectural enhancements would make the contribution analysis more convincing.

- The efficiency discussion also needs to be framed more carefully. The model is clearly parameter-efficient, but the reported GFLOPs and latency are much higher than YOLOv12n, so the paper should **distinguish parameter efficiency from computational efficiency**.  Finally, the numerical inconsistencies around Table 7 should be corrected.

Overall, I think the central performance claim is supported, but some claims about the independent role of TOA-MoE, efficiency, comparison fairness, and result reporting need clearer evidence or more careful wording.

**Requested Changes:**

1. The paper should better separate the effects of the general architectural enhancements from the effects of the proposed TOA-MoE module. Table 4 provides useful ablations for several TOA-MoE components, but TOE-YOLO still groups multiple architectural changes together, including the P2 head, CARAFE, DCNv3, and the feature fusion design. Since TOE-YOLO accounts for a large portion of the improvement over YOLOv12n, a finer ablation would make the contribution analysis more convincing and clarify the incremental value of TOA-MoE.
2. The comparison protocol with external baselines should be clarified. Please specify which baseline results are reproduced and which are taken from prior work, and whether the same data splits, input resolution, training schedule, augmentation strategy, and evaluation protocol are used. This is important for substantiating the SOTA claims.

3. The model is clearly parameter-efficient, but its GFLOPs and latency are substantially higher than YOLOv12n. The paper should explicitly discuss this trade-off and avoid using “lightweight” or “efficient” in a way that only reflects parameter count.

4. The authors should check the numerical inconsistencies around Table 7. For example, the text reports 64.1% overall mAP\@0.5:0.95 on BCCD, while Table 7 reports 63.5; it also reports 48.8% for TOAM-YOLO platelets and 44.9 for YOLOv12n platelets, while Table 7 reports 51.1 and 47.3, respectively.

5.  Since the paper claims complementary specialization of the Fourier and Hessian experts, it would be helpful to include more systematic evidence beyond the qualitative visualization, such as routing-weight statistics, expert contribution analysis, or performance breakdowns under different tiny-object conditions.

---

> ### Author Response · Authors · 2026-06-01
> **Response to Reviewer u7Cs (1/3)**
>
> We thank the Reviewer for their constructive feedback and insightful suggestions. We have carefully addressed each point to improve the technical depth and presentation of the manuscript.
>
> RC: Requested Change
>
> >1.  The paper should better separate the effects of the general architectural enhancements from the effects of the proposed TOA-MoE module. Table 4 provides useful ablations for several TOA-MoE components, but TOE-YOLO still groups multiple architectural changes together, including the P2 head, CARAFE, DCNv3, and the feature fusion design. Since TOE-YOLO accounts for a large portion of the improvement over YOLOv12n, a finer ablation would make the contribution analysis more convincing and clarify the incremental value of TOA-MoE.
>
> RC1: We thank the reviewer for requesting a finer-grained ablation analysis. As observed in Table A1 below (added as Table 4 to the revised manuscript), introduction of the P2-head, meant specifically to address tiny-object detection, indeed provides a substantial improvement over the baseline YOLOv12n (+9.6% mAP50 and +4.94% mAP50-95). Replacing the standard upsampling operation with CARAFE, although improves the performance slightly, but reduces the parameter count, suggesting that CARAFE provides a more effective feature reconstruction mechanism for tiny-object representations. The subsequent incorporation of DCNv3 leads to an additional performance gain, indicating that adaptive spatial sampling further enhances feature extraction. While individual ablations of CARAFE and DCNv3 appear less consequential, removing these modules from the final TOAM-YOLO results in a performance degradation from 68.18% to 66.56% mAP50 and from 29.16% to 27.92% mAP50-95, indicating that they contribute positively to the overall feature representation and complement the proposed TOA-MoE module. Consequently, we retain CARAFE and DCNv3 in the final model configuration. These details are added to the revised draft in blue color in Section 4.2.1, para 3. All rows in this sweep, including the baseline, share one fixed batch size of 16, so that each delta isolates the added module in the new experiments in Table A1. Absolute values thus differ marginally from Tables 3 and 4.
>
> **Table A1: Progressive architectural ablation (SeaPerson Dataset)**
>
> | Model Configuration                       | test mAP50 | test mAP50-95 | Params (M) | GFLOPs   |
> | ----------------------------------------- | ---------- | ------------- | ---------- | -------- |
> | YOLOv12n                                  | 56.43%      | 22.41%         | 2.51       | 5.8      |
> | YOLOv12n + P2 head                        | 66.03%     | 27.35%        | 3.07       | 13.1     |
> | YOLOv12n + p2 + carafe                    | 66.19%     | 27.40%        | 2.83       | 13       |
> | YOLOv12n + P2 + CARAFE + DCNv3 (TOE-YOLO) | 66.79%     | 27.65%        | 2.88       | 13.1     |
> | **TOAM-YOLO (TOE-YOLO + TOA-MoE)**    | **68.18%** | **29.16%**    | **3.3**   | **31.9** |
> | TOAM-YOLO without CARAFE and DCNv3        | 66.56%     | 27.92%        | 3.1        | 31.1     |
>
> > 2. The comparison protocol with external baselines should be clarified. Please specify which baseline results are reproduced and which are taken from prior work, and whether the same data splits, input resolution, training schedule, augmentation strategy, and evaluation protocol are used. This is important for substantiating the SOTA claims.
>
> RC2. We thank the reviewer for pointing this out. In the revised manuscript, we explicitly state which baseline results are reproduced by us and which ones are directly reported from prior works in Section 4.1.4 and in footnote of Table 5 of the revised manuscript. The official dataset splits were used to ensure fair comparison with prior work. Required changes are made to the revised manuscript in the last line of Section 4.1.1 and highlighted in blue color.
>
> The augmentation strategy was kept consistent across the reproduced baselines and the proposed method for each dataset. Default augmentations were used for SeaPerson, TinyPerson, and VisDrone, while dataset-specific augmentations were applied for BCCD and CBC, as summarized in Tables 15 (Appendix B).
>
> Training schedules were kept consistent across experiments. The hyperparameters used for training the CST-YOLO baseline have been added to Table 16 (Appendix B) of the revised manuscript to improve transparency and reproducibility.

---

> > ### Author Response · Authors · 2026-06-01
> > **Response to Reviewer u7Cs (2/3)**
> >
> > > 3. The model is clearly parameter-efficient, but its GFLOPs and latency are substantially higher than YOLOv12n. The paper should explicitly discuss this trade-off and avoid using “lightweight” or “efficient” in a way that only reflects parameter count.
> >
> > RC3. We thank the reviewer for pointing this out. We have revised the manuscript to clearly distinguish parameter efficiency from computational efficiency. Our design prioritizes a small parameter footprint (3.3M parameters), which reduces model storage and memory requirements for edge-oriented deployments, accepting an increase in computational cost as a trade-off for improved feature representation and detection accuracy. To avoid ambiguity, we have revised the manuscript to ensure that terms such as “efficient” and “lightweight” are not used to imply computational efficiency. For further clarity, we have explicitly added the following line in the Introduction (page 2, paragraph 2, last lines highlighted in blue text):
> >
> >
> > “Figure 1b illustrates the trade-off between computational complexity (GFLOPs) and detection accuracy relative to the YOLOv12n baseline. While the proposed approach incurs a higher computational cost, this increase is compensated by a marked improvement in detection accuracy, especially for tiny objects that are often inadequately captured by lightweight architectures.”
> >
> >
> > To contextualize this trade-off, we also report measured inference speed and latency. As shown in Table A2 below (Refer to Table 8 of the revised manuscript), TOAM-YOLO runs at 62.36 FPS (16.04 ms) compared to 89.91 FPS (11.12 ms) for YOLOv12n. These results explicitly quantify the additional computational cost associated with the proposed architecture while demonstrating that the resulting gains in detection accuracy are achieved, while maintaining real-time performance. We have updated the Introduction and Results sections to discuss the parameter, GFLOPs, and accuracy trade-off explicitly. We have removed the words "lightweight" or "efficient" in the context of computational complexity. However, the terms may have been used with the gating mechanism.
> >
> > **Table A2: Inference speed and latency across models**
> >
> > | Model     | FPS   | Inference Time (ms) | mAP50 |
> > | --------- | ----- | ------------------- | ----- |
> > | YOLOv12n  | 89.91 | 11.12               | 56.3 |
> > | YOLOv12s  | 87.80 | 11.39               | 58.40 |
> > | YOLOv12m  | 76.94 | 13.00               | 60.60 |
> > | TOE-YOLO  | 76.11 | 13.14               | 65.2 |
> > | TOAM-YOLO | 62.36 | 16.04               | 67.9 |
> >
> > > 4. The authors should check the numerical inconsistencies around Table 7. For example, the text reports 64.1% overall mAP@0.5:0.95 on BCCD, while Table 7 reports 63.5; it also reports 48.8% for TOAM-YOLO platelets and 44.9 for YOLOv12n platelets, while Table 7 reports 51.1 and 47.3, respectively.
> >
> > RC4. We apologize for these errors and thank the reviewer for catching them. Results reported in Table 7 were all correct. A few results from initial experiments were added to the text that were not updated as the Tables were finalized. We regret this error and have thoroughly proof read the revised draft and checked that there are no inconsistencies in the reported results in Tables and the corresponding text.

---

> > > ### Author Response · Authors · 2026-06-01
> > > **Response to Reviewer u7Cs (3/3)**
> > >
> > > > 5. Since the paper claims complementary specialization of the Fourier and Hessian experts, it would be helpful to include more systematic evidence beyond the qualitative visualization, such as routing-weight statistics, expert contribution analysis, or performance breakdowns under different tiny-object conditions.
> > >
> > > RC5. We thank the reviewer for this valuable suggestion. In response, we conducted additional analyses to quantitatively examine the specialization behavior of the Spatial, Fourier, and Hessian experts within TOA-MoE. A new Section 4.2.2 has been added to the revised manuscript, where we report the mean, standard deviation, minimum, and maximum routing weights generated by the gating network across the evaluation splits of all datasets.
> > > As shown in Table A3 (Table 12 in the revised manuscript), the Hessian expert consistently receives the highest average routing weight across datasets, suggesting that second-order structural information plays an important role during inference. However, the relative contributions of the Spatial and Fourier experts vary across datasets. For example, the Spatial expert receives greater average weight than the Fourier expert on SeaPerson and BCCD, whereas the Fourier expert becomes the second most utilized branch on VisDrone. These trends indicate that expert utilization adapts to dataset characteristics rather than following a fixed preference.
> > > The routing statistics also show that the model does not collapse to a single expert. On SeaPerson, for instance, the Hessian expert's routing weight ranges from 28.42% to 76.40%, while the Spatial expert receives up to 51.83% of the routing weight for certain samples. Overall, these results provide quantitative evidence that the gating network dynamically combines structural, spatial, and frequency-domain representations according to the input data, supporting the intended specialization behavior of the proposed TOA-MoE framework. This analysis and Table A3 have been added to the revised manuscript.
> > >
> > > **Table A3: Routing-weight statistics per expert and dataset (percentages).**
> > >
> > > | Dataset | Expert | Mean | Std | Min | Max |
> > > |---|---|---|---|---|---|
> > > | SeaPerson | Spatial | 20.98 | 5.88 | 18.55 | 51.83 |
> > > | SeaPerson | Fourier | 8.77 | 2.39 | 4.44 | 19.75 |
> > > | SeaPerson | Hessian | 70.26 | 7.80 | 28.42 | 76.40 |
> > > | VisDrone | Spatial | 8.26 | 0.59 | 7.36 | 11.16 |
> > > | VisDrone | Fourier | 11.19 | 1.44 | 7.69 | 14.50 |
> > > | VisDrone | Hessian | 80.55 | 1.72 | 74.34 | 83.86 |
> > > | BCCD | Spatial | 13.73 | 6.71 | 0.53 | 19.28 |
> > > | BCCD | Fourier | 3.74 | 2.09 | 0.14 | 5.40 |
> > > | BCCD | Hessian | 82.54 | 8.77 | 75.67 | 99.32 |
> > >
> > > These routing statistics are also consistent with the component ablations already reported in Table 3 of the manuscript. Although the Fourier expert receives a comparatively small average routing weight, removing it still reduces performance (w/o Fourier: 66.8% mAP50 versus 67.9% for the full model), and removing the Hessian expert produces a larger drop (w/o Hessian: 65.5% mAP50). Removing both experts together lowers performance further (w/o Fourier + Hessian: 64.9% mAP50).The routing-weight statistics and ablation results are therefore mutually consistent: the Hessian expert contributes the largest overall impact, while the Fourier and Spatial experts provide additional non-redundant information that improves detection performance despite receiving lower average routing weights.
> > >
> > > > 6. The paper does not raise major ethical concerns. Still, since the method is evaluated on maritime and drone imagery, it may be useful to briefly acknowledge potential privacy or surveillance-related concerns in a broader impact statement. For the medical blood cell detection experiments, the paper should also clarify that the model is not clinically validated and should not be used as a standalone diagnostic system
> > >
> > > RC6. We thank the reviewer for this suggestion. We have added a Broader Impact statement to the revised manuscript.

---

### Review · Reviewer_zRP4 · 2026-05-19

**Summary Of Contributions:**

This paper proposes **TOAM-YOLO**, a lightweight tiny-object detection framework built on YOLOv12. The primary contribution is the proposed **TOA-MoE** module, which combines spatial, frequency-domain, and Hessian-curvature experts with hierarchical attention operating at pixel, patch, and region levels. The framework further integrates CARAFE upsampling, BiFPN-style feature fusion, DCNv3 modules, and a high-resolution P2 detection head to improve fine-grained feature preservation and tiny-object localization.

Experiments on five datasets (SeaPerson, TinyPerson, VisDrone, BCCD, and CBC) demonstrate consistent improvements over YOLO baselines and prior tiny-object detectors while maintaining a relatively lightweight model size and real-time inference capability. The paper also provides extensive ablation studies and cross-backbone experiments on YOLOv11.

**Additional Comments:**

N/A

**Audience:**

Yes

**Audience Explanation:**

Yes. Object detection is a large and highly active research area, and tiny object detection remains a particularly challenging and practically important subproblem

**Claims And Evidence:**

Yes

**Claims Explanation:**

### Strengths
- **Strong empirical performance across multiple domains.**
  The proposed method consistently improves performance on diverse tiny-object detection benchmarks spanning maritime surveillance, aerial imagery, and medical imaging.

- **Comprehensive ablation studies and statistical analysis.**
  The paper includes detailed ablations for the Fourier and Hessian experts, hierarchical attention levels, DCNv3 modules, and the overall TOA-MoE design.

- **Lightweight design with real-time capability.**
  Despite introducing multiple architectural enhancements, the model remains relatively lightweight (~3.3M parameters) and maintains real-time inference speed, making it potentially practical for deployment-oriented applications.

- **Cross-backbone generalization to YOLOv11.**
  The proposed modules are additionally evaluated on YOLOv11 and demonstrate similarly strong improvements, suggesting that the approach is not tightly coupled to a single YOLO architecture and may generalize across detector backbones.

### Weaknesses
- **Combines many existing components, limiting methodological novelty.**
  The framework integrates several established techniques, including CARAFE, BiFPN-style fusion, DCNv3, FFT-based processing, and attention modules. While the integration is effective, the contribution is primarily an architectural combination and engineering refinement rather than a fundamentally new detection paradigm.

- **Large GFLOPs increase despite small parameter growth.**
  Although the parameter increase is modest, the computational cost rises substantially compared to YOLOv12n. This weakens some efficiency claims and raises concerns regarding deployment on resource-constrained hardware.

- **Limited comparison with recent lightweight transformer-based detectors.**
  The evaluation mainly compares against YOLO-family models and a few prior tiny-object detectors such as TPS-YOLO and DDH-YOLO. While these are reasonable baselines, recent lightweight transformer-based or hybrid CNN-transformer detectors have shown strong performance on small-object and dense-scene detection tasks. The paper briefly discusses DETR-style approaches in related work, but experimental comparisons remain limited. Since one of the central claims is that TOAM-YOLO achieves state-of-the-art tiny-object detection performance while remaining efficient, comparisons against stronger modern lightweight transformer/hybrid detectors would make the claims substantially more convincing and better position the work within the current literature.

- **Over-specialized evaluation.**
  The evaluation is primarily conducted on specialized tiny-object datasets (SeaPerson, TinyPerson, VisDrone) and relatively small medical datasets (BCCD, CBC), which limits the breadth of empirical validation. While these benchmarks are relevant to the target application, the absence of experiments on widely adopted large-scale object detection benchmarks such as COCO, Objects365, or PASCAL VOC makes it difficult to assess the generalizability of the proposed architecture beyond highly specialized tiny-object scenarios.

- **Missing details.**
How to generate these weights {$w_{id}, w_{freq}, w_{hess}$}  is not clear.

**Requested Changes:**

Please address the comments mentioned above.

In addition, there are several issues regarding the paper presentation and writing quality.

For example, on page 16, there is a large blank space and a noticeable formatting gap between sections and subsections, which negatively affects readability and the manuscript's overall presentation quality.

Some paragraphs are extremely long, spanning multiple lines.

The paper would benefit from a more careful proofreading and formatting pass to improve clarity, consistency, and professionalism.

---

> ### Author Response · Authors · 2026-06-01
> **Response to Reviewer zRP4 (1/3)**
>
> We thank the Reviewer for their constructive feedback and insightful suggestions. We have carefully addressed each point to improve the technical depth and presentation of the manuscript.
>
> RC: Requested Change
>
> > 1. The framework integrates several established techniques, including CARAFE, BiFPN-style fusion, DCNv3, FFT-based processing, and attention modules. While the integration is effective, the contribution is primarily an architectural combination and engineering refinement rather than a fundamentally new detection paradigm.
>
> RC1. We thank the reviewer for this observation and agree that several individual components employed in the framework, such as CARAFE, DCNv3, and BiFPN-style fusion, are established techniques. Our contribution is, therefore, not the introduction of a fundamentally new detection paradigm, but the design of a unified architecture specifically tailored to mitigate information loss in tiny-object detection.
> In particular, the proposed TOAM module introduces a mixture-of-experts framework that combines spatial, frequency-domain, and Hessian-based representations through adaptive routing, enabling the network to dynamically emphasize complementary feature types. To the best of our knowledge, this combination has not been explored previously in this form for tiny-object detection.
> The accompanying ablation studies demonstrate that the observed performance gains arise from the interaction of these components rather than from any single module or a simple increase in model capacity. Furthermore, the consistent improvements obtained across multiple tiny-object benchmarks spanning aerial, maritime, and medical imaging domains, as well as across both YOLOv11 and YOLOv12 backbones, suggest that the proposed design captures broadly applicable principles for enhancing tiny-object detection performance. We have revised the manuscript to more clearly distinguish the novelty of the proposed architectural integration from the individual techniques on which it is built.
> > 2. Although the parameter increase is modest, the computational cost rises substantially compared to YOLOv12n. This weakens some efficiency claims and raises concerns regarding deployment on resource-constrained hardware.
>
> RC2. We thank the reviewer for pointing this out. Our design prioritizes a small parameter footprint (3.3M parameters), which reduces model storage and memory requirements for edge-oriented deployments, accepting an increase in computational cost as a trade-off for improved feature representation and detection accuracy. For further clarity, we have explicitly added the following line in the Introduction (page 2, paragraph 2, last lines highlighted in blue text):
>
> “Figure 1b illustrates the trade-off between computational complexity (GFLOPs) and detection accuracy relative to the YOLOv12n baseline. While the proposed approach incurs a higher computational cost, this increase is compensated by a marked improvement in detection accuracy, especially for tiny objects that are often inadequately captured by lightweight architectures.”
>
> To contextualize this trade-off, we also report measured inference speed and latency. As shown in Table A1 below (Refer to Table-8 of the revised manuscript), TOAM-YOLO runs at 62.36 FPS (16.04 ms) compared to 89.91 FPS (11.12 ms) for YOLOv12n. These results explicitly quantify the additional computational cost associated with the proposed architecture while demonstrating that the resulting gains in detection accuracy are achieved, while maintaining real-time performance. We have updated the Introduction and Results sections to discuss the parameter, GFLOPs, and accuracy trade-off explicitly.
>
> **Table A1: Inference speed and latency across models**
>
> | Model     | FPS   | Inference Time (ms) | mAP50 |
> | --------- | ----- | ------------------- | ----- |
> | YOLOv12n  | 89.91 | 11.12               | 56.3 |
> | YOLOv12s  | 87.80 | 11.39               | 58.40 |
> | YOLOv12m  | 76.94 | 13.00               | 60.60 |
> | TOE-YOLO  | 76.11 | 13.14               | 65.2 |
> | TOAM-YOLO | 62.36 | 16.04               | 67.9 |

---

> ### Author Response · Authors · 2026-06-01
> **Response to Reviewer zRP4 (2/3)**
>
> > 3. The evaluation mainly compares against YOLO-family models and a few prior tiny-object detectors such as TPS-YOLO and DDH-YOLO. While these are reasonable baselines, recent lightweight transformer-based or hybrid CNN-transformer detectors have shown strong performance on small-object and dense-scene detection tasks. The paper briefly discusses DETR-style approaches in related work, but experimental comparisons remain limited. Since one of the central claims is that TOAM-YOLO achieves state-of-the-art tiny-object detection performance while remaining efficient, comparisons against stronger modern lightweight transformer/hybrid detectors would make the claims substantially more convincing and better position the work within the current literature.
>
> RC3. We thank the reviewer for this valuable suggestion. To address this concern, we included an additional comparison with RT-DETR-L, a modern transformer-based detector, trained using the Ultralytics framework for 200 epochs under the same experimental setting. The results are reported in Table A2 below (Table 7 in the revised manuscript).
>
> **Table A2: Comparison with the transformer-based RT-DETR-L detector on the SeaPerson dataset.**
>
> | Method           | Params | GFLOPs | mAP\@0.5 | mAP\@0.95 |
> | ---------------- | ------ | ------ | -------- | --------- |
> | RT-DETR-L        | 32.81M | 108    | 70.87    | 29.50     |
> | TOAM-YOLO (v11n) | 3.34M  | 32.4   | 69.52    | 29.76     |
> | YOLOv11n         | 2.59M  | 6.4    | 58.99    | 24.36     |
> | TOAM-YOLO (v12n) | 3.3M   | 32     | 67.90    | 28.92     |
> | YOLOv12n         | 2.51M  | 5.8    | 56.3    | 22.7     |
>
> TOAM-YOLO (v11n) achieves competitive performance relative to RT-DETR-L, obtaining 69.52\% mAP@0.5 compared to 70.87\% for RT-DETR-L, while achieving a slightly higher mAP@0.95 of 29.76\% versus 29.50\%. Importantly, this performance is obtained with only 3.34M parameters and 32.4 GFLOPs, compared with 32.81M parameters and 108 GFLOPs for RT-DETR-L. Similarly, TOAM-YOLO (v12n) substantially narrows the performance gap relative to its YOLOv12n baseline while maintaining a compact model size.
>
> These results suggest that the proposed TOAM modules enable lightweight CNN-based detectors to achieve accuracy that is competitive with a substantially larger transformer-based model, while requiring approximately 10x fewer parameters and 3.3x lower computational cost. The comparison therefore highlights the favorable accuracy-efficiency trade-off achieved by the proposed framework for tiny-object detection.
>
> Additionally, we have revised the manuscript to moderate the original state-of-the-art claims and instead emphasize the plug-and-play capability of the proposed modules, their consistent performance gains across multiple architectures and datasets, and the resulting accuracy-efficiency trade-off throughout the manuscript.

---

> > ### Author Response · Authors · 2026-06-01
> > **Response to Reviewer zRP4 (3/3)**
> >
> > > 4. The evaluation is primarily conducted on specialized tiny-object datasets (SeaPerson, TinyPerson, VisDrone) and relatively small medical datasets (BCCD, CBC), which limits the breadth of empirical validation. While these benchmarks are relevant to the target application, the absence of experiments on widely adopted large-scale object detection benchmarks such as COCO, Objects365, or PASCAL VOC makes it difficult to assess the generalizability of the proposed architecture beyond highly specialized tiny-object scenarios.
> >
> > RC4. We thank the reviewer for highlighting the importance of broader empirical validation beyond specialized tiny-object benchmarks. We agree that demonstrating performance on a large-scale generic object detection benchmark provides a stronger assessment of the generalizability of the proposed architecture.
> > To address this concern, we conducted additional experiments on the MS-COCO dataset. Specifically, YOLOv12n and TOAM-YOLO were trained using identical hyperparameters for 100 epochs and evaluated on the COCO val2017 validation set. The results are summarized in Table A3 below (Table 11 in the revised manuscript).
> >
> > **Table A3: Generalization evaluation of TOAM-YOLO and YOLOv12n on the MS-COCO val2017 benchmark trained for 100 epochs.**
> >
> > | Method                 | AP50  | AP50:95 | AP75  | AP_Small | AP_Med | AP_Large | AR_Small | AR_Med | AR_Large |
> > |------------------------|--------|----------|--------|----------|--------|----------|----------|--------|----------|
> > | YOLOv12n               | 48.90% | 34.20%   | 36.80% | 16.50%   | 37.60% | 50.40%   | 34.00%   | 63.80% | 76.70%   |
> > | TOAM-YOLO (v12n based) | 52.40% | 37.50%   | 40.90% | 20.40%   | 39.40% | 50.80%   | 41.30%   | 66.30% | 77.10%   |
> >
> > The COCO experiments show that the proposed TOAM modules consistently improve detection performance across all object scales. Relative to YOLOv12n, TOAM-YOLO improves AP50 by 3.5% (48.9% → 52.4%) and AP50:95 by 3.3% (34.2% → 37.5%). Notably, the largest gains are observed for small objects, with APSmall increasing by 3.9 percentage points (16.5% → 20.4%) and AR_Small by 7.3 percentage points (34.0% → 41.3%). Improvements are also observed for medium-sized objects (AP_Med: +1.8%, AR_Med: +2.5%) and large objects (AP_Large: +0.4%, AR_Large: +0.4%).
> > These results indicate that the benefits of TOAM are not confined to specialized tiny-object datasets and extend to a large-scale, diverse detection benchmark. At the same time, the substantially larger gains observed for small objects are consistent with the design objective of TOAM, namely the preservation and enhancement of fine-grained feature representations that are particularly important for challenging small-object detection scenarios. We have incorporated these additional experiments and discussion into the revised manuscript to clarify both the generalizability of the proposed framework and its particular effectiveness for small-object detection.
> >
> > > 5. Missing details. How to generate these weights {$w_{id}, w_{freq}, w_{hess}$} is not clear.
> >
> > RC5. We thank the reviewer for pointing out that the mathematical formulation of the fusion weight generation process ($w_{id}$, $w_{freq}$, $w_{hess}$) was not sufficiently explicit in the manuscript. While the original text stated that the weights are produced by a lightweight gating network using adaptive pooling, convolutional layers, and softmax normalization, we have revised the Method Section no. 3.2.1 to explicitly present the routing logits, softmax-based weight computation, and the resulting fusion formulation.
> >
> > Specifically, the TOAM module first generates three feature representations from the same input feature map: the identity feature ($F_{id}$), the frequency-domain feature ($F_{freq}$), and the Hessian-enhanced feature ($F_{hess}$). A lightweight routing branch then processes the input feature map and produces three routing logits:
> >
> > $$[z_{id}, z_{freq}, z_{hess}] = R(X),$$
> >
> > where $R(\cdot)$ denotes the learnable routing network. The logits are normalized using a softmax function to obtain the fusion weights:
> >
> > $$w_i=\frac{e^{z_i}}{\sum_j e^{z_j}}, \quad i\in{id,freq,hess}.$$
> >
> > The final fused representation is computed as:
> >
> > $$F_{moe}=w_{id}F_{id}+w_{freq}F_{freq}+w_{hess}F_{hess}.$$
> >
> > The routing network and fusion weights are optimized jointly during end-to-end training, allowing the contribution of each branch to be adaptively determined for each input feature map.
> >
> > > Paper Presentation and Writing Quality
> >
> > RC5. We thank the reviewer for highlighting these presentation and formatting issues. We have carefully reviewed the manuscript and corrected the formatting inconsistencies, including the excessive whitespace identified on page 16 and spacing irregularities between sections and subsections. In addition, some overly long paragraphs in section 4.2 have been revised and reorganized to improve readability and clarity.

---

### Author Response · Authors · 2026-06-01
**Summary of Revisions**

We sincerely thank all the reviewers for their careful evaluation of our manuscript, constructive feedback, and insightful suggestions. We greatly appreciate the time and effort invested in reviewing our work. The comments have helped us improve both the technical quality and presentation of the manuscript.

We have carefully considered all reviewer comments and have revised the manuscript accordingly. The revised version includes additional experiments, analyses, clarifications, and improvements to the technical depth and presentation of the manuscript. **For ease of review, all modifications made in response to the reviewers' comments are highlighted in blue throughout the revised manuscript.**

---

### Decision · Action_Editor_6ULa · 2026-06-23

**Recommendation:** Accept with minor revision

**Additional Comments:**

The final version should be updated according to reviewers' comments. For example, Reviewer u7Cs mentioned their remaining concern is mainly about claim calibration: the final manuscript should clearly state that the largest gain comes from the P2 pathway, while TOA-MoE provides an additional improvement on top of TOE-YOLO.

**Audience:**

Yes

**Audience Explanation:**

This paper focuses on the object detection task, a common application in machine learning and computer vision.

**Claims And Evidence:**

Yes

**Claims Explanation:**

This paper provides extensive experimental results. It systematically demonstrates the plug-and-play capability of these changes on YOLOv11 and YOLOv12 models. Tiny object aware Mixture of experts based YOLO (TOAM-YOLO) achieves consistent improvements across five datasets: three tiny object benchmarks (SeaPerson, TinyPerson, VisDrone) with mAP@0.5 improvements of 11.6%, 4.64%, and 11% respectively, and two blood cell datasets (BCCD, CBC) with mAP@0.5:0.95 improvements of 3.9% and 1.7% for platelet detection, all over YOLOv12n, while adding only 0.75M parameters.

---

> ### Author Response · Authors · 2026-07-06
> **Camera-Ready Submission**
>
> We thank the Action Editor and the reviewers for their valuable feedback, which helped improve the quality and clarity of our work.
>
> The submitted camera-ready version incorporates all revisions made in response to the reviewers' comments. In particular, as highlighted by the Action Editor regarding Reviewer u7Cs' comment on claim calibration, the manuscript clarifies that the high-resolution P2-level detection pathway provides the largest performance improvement, while TOA-MoE provides an additional improvement on top of TOE-YOLO. This clarification is reflected in the abstract, Section 3.1, Section 4.2.1, the discussion, and the conclusion.
>
> We have also incorporated all other reviewer suggestions into the final camera-ready manuscript. We sincerely thank the Action Editor and the reviewers for their constructive feedback.